# *Tmsb10* triggers fetal Leydig differentiation by suppressing the RAS/ERK pathway

Miki Inoue [1,2], Takashi Baba[1,2], Fumiya Takahashi[1], Miho Terao[3], Shogo Yanai[1], Yuichi Shima[1,2], Daisuke Saito[1,4], Kei Sugihara [5], Takashi Miura[5], Shuji Takada [3], Mikita Suyama [1,4], Yasuyuki Ohkawa [1,6] & Ken-ichirou Morohashi[1,2 ✉]

Leydig cells in fetal testes play crucial roles in masculinizing fetuses through androgen production. Gene knockout studies have revealed that growth factors are implicated in fetal Leydig cell (FLC) differentiation, but little is known about the mechanisms regulating this process. We investigate this issue by characterizing FLC progenitor cells using single-cell RNA sequencing. The sequence datasets suggest that *thymosin β10* (*Tmsb10*) is transiently upregulated in the progenitors. While studying the function of *Tmsb10*, we reveal that platelet-derived growth factor (PDGF) regulates ciliogenesis through the RAS/ERK and PI3K/AKT pathways, and thereby promotes desert hedgehog (DHH)-dependent FLC differentiation. *Tmsb10* expressed in the progenitor cells induces their differentiation into FLCs by suppressing the RAS/ERK pathway. Through characterizing the transiently expressed *Tmsb10* in the FLC progenitors, this study unveils the molecular process of FLC differentiation and shows that it is cooperatively induced by DHH and PDGF.

[1] Department of Systems Life Sciences, Graduate School of Systems Life Sciences, Kyushu University, Maidashi 3-1-1, Higashi-ku, Fukuoka 812-8582, Japan. [2] Department of Molecular Biology, Graduate School of Medical Sciences, Kyushu University, Maidashi 3-1-1, Higashi-ku, Fukuoka 812-8582, Japan. [3] Department of Systems BioMedicine, National Research Institute for Child Health and Development, 2-10-1 Okura, Setagaya-ku, Tokyo 157-8535, Japan. [4] Division of Bioinformatics, Medical Institute of Bioregulation, Kyushu University, Maidashi 3-1-1, Higashi-ku, Fukuoka 812-8582, Japan. [5] Department of Anatomy and Cell Biology, Graduate School of Medical Sciences, Kyushu University, Maidashi 3-1-1, Higashi-ku, Fukuoka 812-8582, Japan. [6] Division of Transcriptomics, Medical Institute of Bioregulation, Kyushu University, Maidashi 3-1-1, Higashi-ku, Fukuoka 812-8582, Japan. ✉email: morohashi.ken-ichirou.874@m.kyushu-u.ac.jp

Two types of somatic cells, Sertoli and Leydig cells, play unique and mutually complementary roles to achieve testicular functions. Sertoli cells provide germ and Leydig cells with nutrients and regulatory molecules to regulate their differentiation and functions, and in turn, Leydig cells produce a potent androgen, testosterone, that regulates the differentiation and functions of both germ and Sertoli cells[1]. In addition to these intratesticular functions, testosterone induces a variety of male traits throughout the body. Unlike other vertebrates, mammalian species possess two types of Leydig cells, fetal-type Leydig cells (FLCs) and adult-type Leydig cells (ALCs), the former of which play pivotal roles in the masculinization of fetuses through androgen production[2–4].

Several studies using gene-disrupted mice have revealed the involvement of multiple growth factors and their receptors in gonad development. Among them, desert hedgehog (DHH), NOTCH, platelet-derived growth factor (PDGF), and transforming growth factor β (TGFβ) signals were implicated in FLC differentiation; disruption of these genes resulted in aberrant FLC differentiation[5–8]. Likewise, genes encoding transcription factors such as *aristaless related homeobox* (*Arx*)[9], *podocyte-expressed 1/transcription factor 21* (*Pod1/Tcf21*)[10], *adrenal 4-binding protein/steroidogenic factor-1* (*Ad4BP/SF-1, Nr5a1*)[11,12], glioma-associated Krüppel-type Zn finger protein (*Gli1/Gli2*)[13], and *Gli3*[14] were found to contribute to FLC differentiation as demonstrated in gene knockout (KO) mice. Moreover, possible FLC progenitor cells have been shown to express ARX[15], MAFB[16], Notch[17,18], Nestin[19], and *Wnt5a*[20,21]. Therefore, several studies have shown that multiple factors are involved in the differentiation of FLCs. However, it remains unclear how these factors result in the formation of FLC progenitor cells and then promote their successive differentiation into FLCs.

Studies concerning hedgehog (HH) signaling have unveiled the complex mechanism of intracellular signal transduction. Upon binding of HH to its receptor Patched (PTCH), Smoothened (SMO) is released from inhibition by the receptor and then accumulates in the primary cilium[22]. Thereafter, SMO undergoes phosphorylation, and in this form it promotes dissociation of GLI from kinesin family protein 7 and Suppressor of fused. Ultimately, GLI is converted to an active form and then begins to transcribe HH target genes[23].

Studies of PDGF signaling have demonstrated that two receptors, PDGFRα and PDGFRβ, transduce signals upon binding to four ligand molecules, PDGF-A, PDGF-B, PDGF-C, and PDGF-D[24]. The ligand-bound receptors activate the RAS/ERK pathway by successive phosphorylation of its components. It has also been shown that RAS activates the PI3K/AKT pathway, in which phosphatidylinositol 3-phosphate (PIP$_3$), which is synthesized by PI3K, plays a pivotal role in activating PDK1 and AKT. By regulating these signal pathways, PDGFs are involved in a variety of cellular processes such as differentiation, proliferation, metabolism, and migration[24]. As described above, the specific mechanisms of DHH and PDGF signal transduction have been uncovered gradually. Unfortunately, however, it remains largely unknown how these growth factors promote FLC differentiation.

Thymosins were originally isolated from the calf thymus[25]. Among them, TMSB4X (thymosin beta 4, X chromosome) and TMSB10 (thymosin beta 10), which are members of the β-thymosin family, have highly homologous amino acid sequences. The expression of these β-thymosins has been observed in a variety of normal and cancer cells[26–28]. As for their functions, TMSB4X was shown to sequester actin monomer to suppress the formation of filamentous actin[29]. Likewise, TMSB10 suppresses actin polymerization through the actin-binding sequence conserved in these β-thymosins[30]. Related to their actin-sequestering function, many studies have reported that these β-thymosins are potentially involved in processes such as blood vessel formation, wound healing, cell migration, and cancer metastasis[31]. Moreover, TMSB10, but not TMSB4X, interacts directly with RAS to inhibit RAS-RAF interaction, which disturbs downstream signal transduction[32].

A transgenic *FLE-EGFP* mouse was established using the fetal Leydig-specific enhancer (FLE) and promoter region of *Ad4BP/SF-1* gene[33]. FLCs in the fetal testes of the transgenic mice were strongly labeled with EGFP (S-EGFP cells), and a large population of non-steroidogenic interstitial cells was labeled only weakly with EGFP (W-EGFP cells). Both S-EGFP and W-EGFP cells could be recovered separately by fluorescence-activated cell sorting (FACS) from E16.5 fetal testes[34]. Because FLCs at this stage increase in number even though they scarcely proliferate[15,19,35], we anticipated that the W-EGFP cell population may include FLC progenitor cells. Indeed, in vitro testis reconstruction studies demonstrated that some W-EGFP cells, if not all, have the potential to differentiate into FLCs[36].

In the present study, we examined the interstitial cells of developing fetal testes in mice using single-cell RNA sequencing (scRNA-seq). Analyses of the sequence datasets found a unique cell fraction potentially consisting of FLC progenitors. Among the genes whose expression was upregulated in the progenitor cells, we focused on *Tmsb10*. We found that PDGF regulated the formation of primary cilia via signaling in the RAS/ERK and PI3K/AKT pathways, and moreover that TMSB10 promoted ciliation by hindering the interaction between RAS and RAF, thereby suppressing the RAS/ERK signal pathway. This study unveils, to the best of our knowledge, part of the molecular process of FLC differentiation that is induced cooperatively by DHH and PDGF.

## Results

**FLC progenitor cells are present in the interstitial space of fetal testes**. To identify the FLC progenitors, 696 W-EGFP cells (a large population of interstitial cells) and 92 S-EGFP cells (FLCs) prepared from E16.5 fetal testes *FLE-EGFP* mouse[33] were subjected to scRNA-seq[37]. After low-quality scRNA-seq datasets were removed, 341 and 80 datasets obtained from the W-EGFP and S-EGFP cells, respectively, were subjected to subsequent analyses (Supplementary Table 1).

To assess how many cell types were present among the W-EGFP cells, the datasets were subjected to hierarchical clustering on the principal components. As indicated in the cluster dendrogram in Fig. 1a, the W-EGFP cells were divided into three clusters (clusters A, B, and C), while the S-EGFP cells were divided into two clusters (clusters D and E). According to the relative distances between the clusters, cluster E was the most distant from the other clusters, while clusters A and B were not clearly segregated. Interestingly, clusters C and D, which were originally derived from the W-EGFP and S-EGFP cells, respectively, demonstrated an intimate correlation. A similar distribution of the five cell clusters was observed by t-distributed stochastic neighboring embedding (t-SNE) (Fig. 1b).

The scRNA-seq data were further subjected to Monocle trajectory analysis to predict the developmental trajectory of the five cell clusters. As indicated in Fig. 1c, the cells in clusters A and B were predicted to differentiate into those in cluster C, and eventually, via cluster D, into those in cluster E. Because the cells in cluster E are FLCs, as described later, characterization of the putative progenitor cells in cluster C seemed to be critical for uncovering the mechanism of FLC differentiation.

Therefore, a heatmap of gene expression was generated using the genes whose expression was altered in clusters C, D, and E (Fig. 1a). The genes in group I showed higher expression in cluster E and/or cluster D. FLC marker genes such as *insulin-like 3* (*Insl3*)[38,39], *cholesterol side-chain cleavage enzyme cytochrome 450* (*Cyp11a1*), and *3β-hydroxysteroid dehydrogenase* (*Hsd3b1*)[40–42] were included

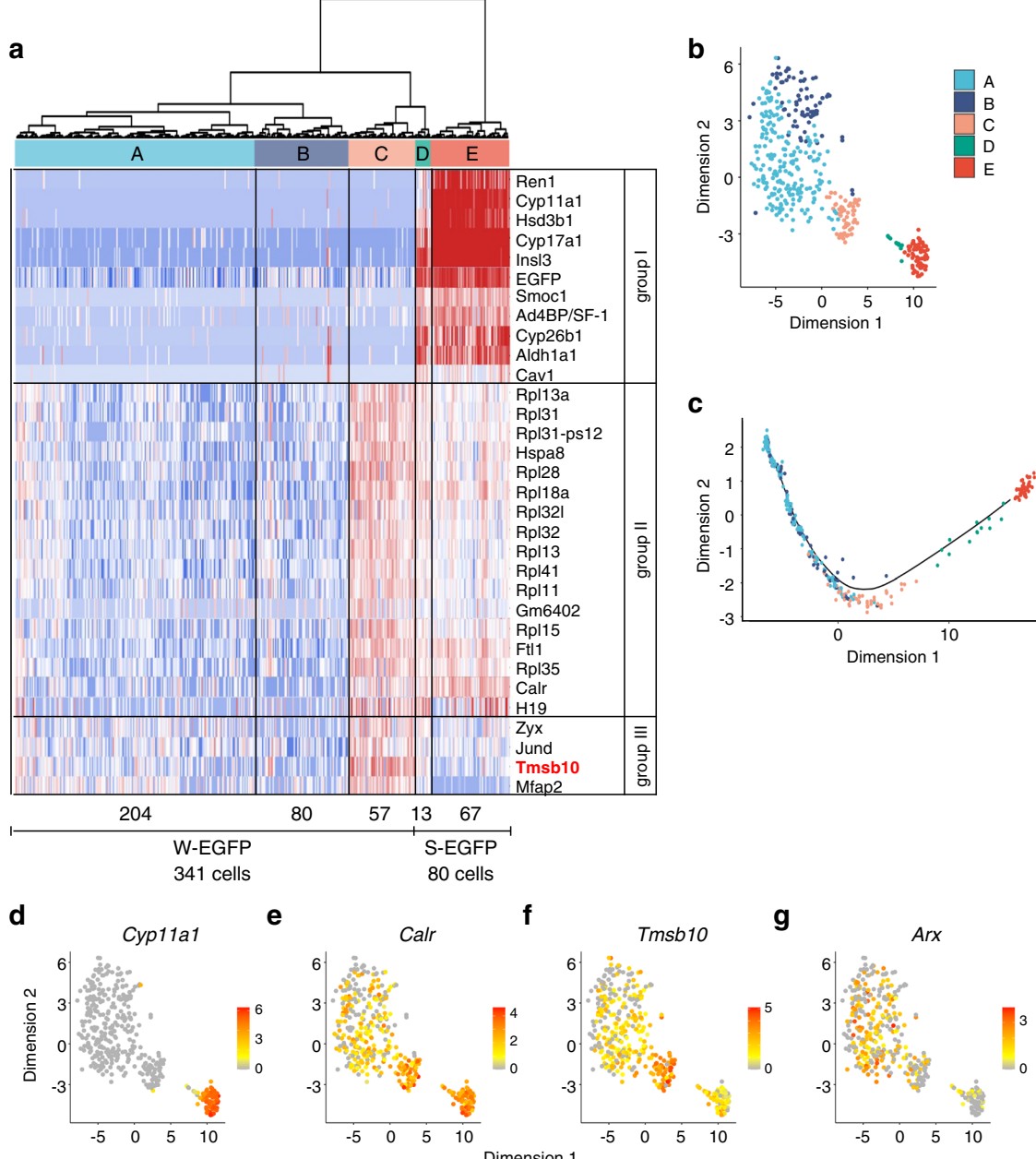

**Fig. 1 Characterization of S- and W-EGFP cells by scRNA-seq. a** High-quality scRNA-seq datasets obtained from 80 S-EGFP and 341 W-EGFP cells were analyzed by hierarchical clustering. As indicated by the dendrogram (upper), the cells were divided into five cell clusters: cluster A (204 W-EGFP cells; light blue), cluster B (80 W-EGFP cells; blue), cluster C (57 W-EGFP cells; orange), cluster D (13 S-EGFP cells; green), and cluster E (67 S-EGFP cells; red). The heatmap (lower) was based on genes in clusters C, D, and/or E whose expressions were altered. Genes differentially expressed in these clusters (group I to III) are shown. *Tmsb10* in group III is highlighted in red. **b**, **c** Results of t-SNE **b** and Monocle pseudo-time trajectory analyses **c** of the datasets are shown. The cellular distribution is shown (each dot represents one cell), with colors labeling the five clusters of cells as in **a**. **d**–**g** Expression levels of *Cyp11a1* **d**, *Calr* **e**, *Tmsb10* **f**, and *Arx* **g** in the cells above, as determined by scRNA-seq. Each dot represents one cell.

in this group, indicating that the cells in clusters D and E consisted of immature and mature FLCs, respectively. The genes in group II showed higher expression in clusters C, D, and E. Interestingly, several genes encoding large ribosomal subunits were included in this group. Finally, the genes in group III showed higher expression in cluster C.

**Expression of Tmsb10 is transiently upregulated in putative FLC progenitors.** Expression levels of the genes above were depicted colorimetrically on the cells whose distribution was determined by t-SNE (Fig. 1d–f, Supplementary Fig. 1a). As

expected, the highest expression of *Cyp11a1* was seen in cluster E, that of *calreticulin* (*Calr*) was in clusters C, D, and E, and that of *Tmsb10* was in cluster C. Considering that the cells in cluster C are FLC progenitors, the expression of *Tmsb10* appeared to be transiently upregulated during differentiation into FLCs. The expression pattern of *Tmsb4x*, another member of β-thymosin family, was different from that of *Tmsb10* (Supplementary Fig. 1b). We previously showed that the expression of ARX is gradually decreased in interstitial cells that are differentiating into FLCs[15]. Consistent with the previous observation, *Arx* expression appeared to be decreased in cluster C and then further decreased in clusters D and E (Fig. 1g).

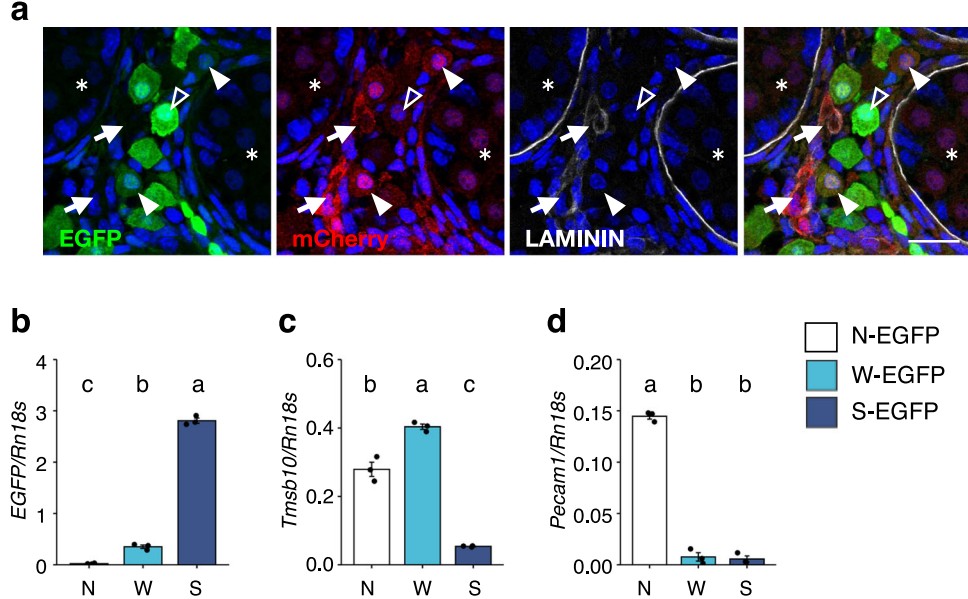

**Fig. 2 Increased expression of *Tmsb10* in putative FLC progenitors. a** The testes of *FLE-EGFP;Tmsb10-mCherry* mouse fetuses at E16.5 were analyzed by immunofluorescence. Representative images of staining with EGFP (green), mCherry (red), and LAMININ (white), with all three stains overlaid in the image on the right. Arrows indicate cells double positive for mCherry and LAMININ. Closed arrowheads indicate cells double positive for mCherry (weakly stained) and EGFP. Open arrowheads indicate FLCs strongly stained with EGFP. Asterisks mark testis tubules. Scale bar = 20 μm. **b–d** Expressions of *EGFP* **b**, *Tmsb10* **c**, and *Pecam1* **d** in N-EGFP (EGFP-negative, N, open bars), W-EGFP (W, light blue bars), and S-EGFP (S, blue bars) cells prepared from *FLE-EGFP* fetal testes at E16.5. The data were normalized by *Rn18s* and are presented as means ± SEM. Letters a, b, and c denote significant differences between the cell groups, N-EGFP (N), W-EGFP (W), and S-EGFP (S). n = 3. p < 0.01.

**TMSB10 is expressed in the interstitial cells of fetal testes.**
Focusing on the gene expression profile of the genes in group III, we investigated whether *Tmsb10* is involved in FLC differentiation. First, we attempted to identify the cells expressing TMSB10 in fetal testes. Because no available antibody recognized TMSB10, we generated knock-in mice in which *mCherry* was inserted at the *Tmsb10* locus (Supplementary Fig. 2). The mice were then crossed with *FLE-EGFP* mice, and then the fetal testes of the double transgenic mice at E16.5 were examined. TMSB10 (mCherry) did not seem to be expressed in FLCs (S-EGFP cells) (open arrowheads in Fig. 2a). Strong signals for TMSB10 were detected in the interstitial space, and they were colocalized with laminin (arrows). Considering that laminin is a marker of endothelial cells (Supplementary Fig. 3), TMSB10 was thought to be strongly expressed in the cells. Further, cells exhibiting weak TMSB10 signals were present in the interstitial space (closed arrowheads), and some of these cells were weakly stained with EGFP.

To exclude the possibility that the W-EGFP cell population includes *Tmsb10*-expressing endothelial cells, we examined the expression of the endothelial cell marker *Pecam1* in W-EGFP, S-EGFP, and EGFP-negative cells. qRT-PCR revealed that *Pecam1*-positive endothelial cells were mostly recovered in EGFP-negative cells but not in W-EGFP cells (Fig. 2b–d), indicating that W-EGFP cells include TMSB10-positive cells other than endothelial cells.

**Tmsb10 is required for FLC differentiation.** As described previously[36], we established a testis reconstruction system using W-EGFP cells mixed with whole cells prepared from wild-type fetal testes (Fig. 3a). Using this system, we succeeded in recapitulating FLC differentiation from W-EGFP cells by detecting the appearance of cells strongly positive for EGFP (equivalent to S-EGFP cells) (Fig. 3b). A previous gene disruption study demonstrated that DHH is required for differentiation of FLCs[5]. Thus, we investigated whether DHH stimulates FLC differentiation using in vitro reconstructed testes. As shown in Fig. 3c, d,

EGFP signals in the reconstructed testes increased when incubated in the presence of a SMO agonist (SAG).

Considering the transiently upregulated expression of *Tmsb10* in the putative progenitor cells, we hypothesized that *Tmsb10* plays a critical role in FLC differentiation. Thus, we examined whether *Tmsb10* knockdown (KD) impacted FLC differentiation in the reconstructed testes. Cultured W-EGFP cells were treated with the siRNAs for *Tmsb10*, *Tmsb4x*, and *Ad4BP/SF-1*. As expected, these treatments resulted in a clear reduction of the expression of each corresponding gene (Supplementary Fig. 4). Then, we utilized these KD cells for in vitro testis reconstruction assays. *Tmsb10* KD was found to impair the ability of W-EGFP cells to differentiate into S-EGFP FLCs, although this impairment was not observed following treatment with si*Tmsb4x* or control siRNA (si*Cnt*) (Fig. 3e). It has been established that *Ad4BP/SF-1* is essential for steroidogenic cell differentiation[11,12,43,44]. As expected, FLC differentiation was markedly affected by *Ad4BP/SF-1* KD (Fig. 3e). Quantitative examination indicated that *Tmsb10* KD decreased the differentiation efficacy to 37% of control, whereas it was unaffected by *Tmsb4x* KD (Fig. 3f). *Ad4BP/SF-1* KD resulted in a reduction of more than 90%. Similar effects of *Tmsb10* KD on FLC differentiation were observed in the presence of SAG (Fig. 3g, h).

Finally, we examined whether activity downstream of HH signaling was affected by *Tmsb10* KD. *Gli1* gene expression is known to be activated by HH signaling[23]. As expected, *Tmsb10* KD resulted in a decrease of *Gli1* expression (Fig. 3i), while *Tmsb4x* KD had no effect (Supplementary Fig. 5a–c). Taken together, these results strongly suggest that *Tmsb10* is required for FLC differentiation through regulating HH signaling.

**Tmsb10 regulates primary cilia formation by inhibiting the RAS/ERK pathway.** Because the primary cilium is a unique structure that is essential for HH signal transduction[22], we examined whether *Tmsb10* was implicated in the regulation of ciliogenesis.

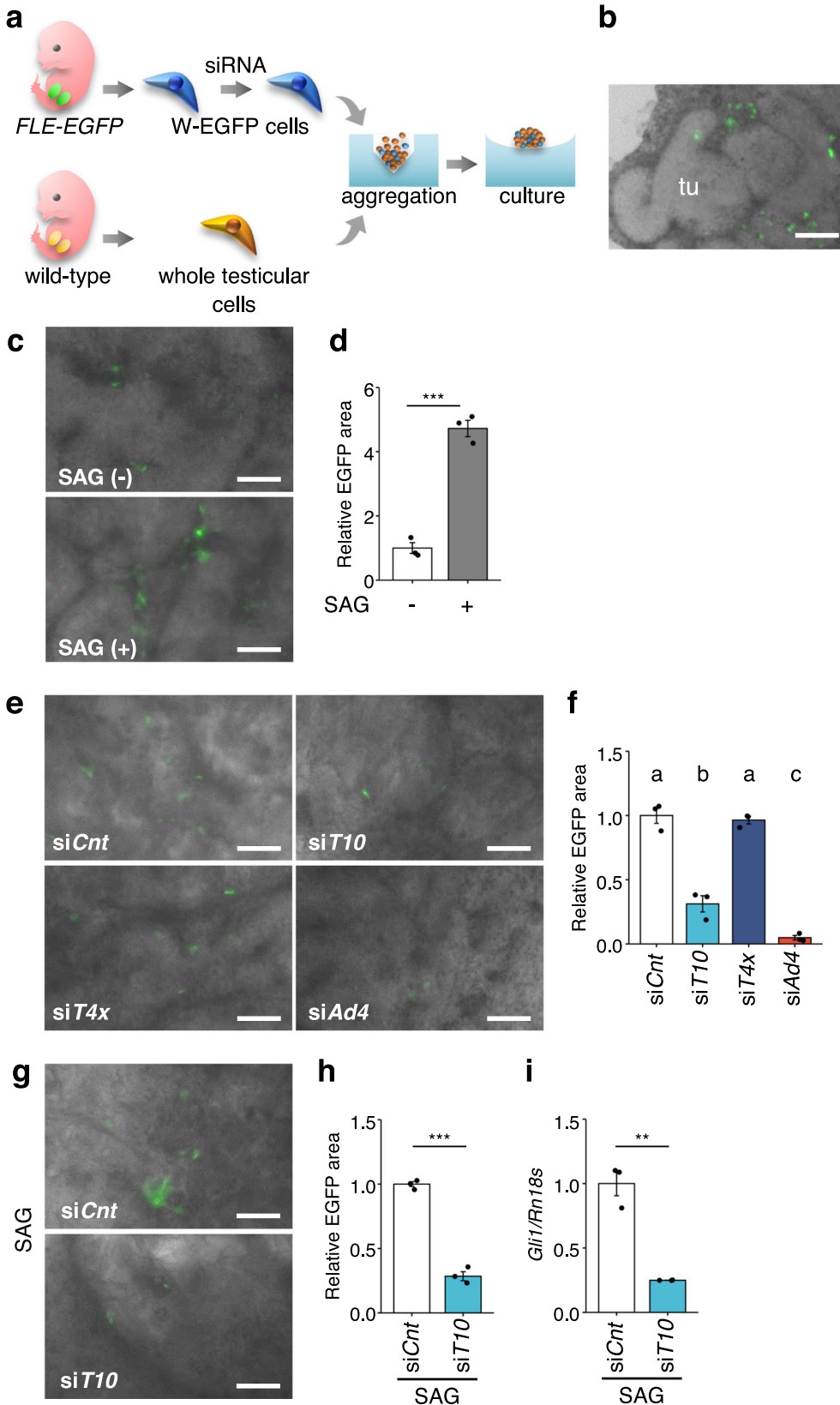

Consistent with the observation that primary cilia emerge only during the G0/G1 cell cycle phase[45], we scarcely detected them in W-EGFP cells cultured in serum-containing medium. Therefore, W-EGFP cells were cultured in serum-free medium for 24 h before the following immunofluorescence studies were performed. Under this condition, approximately 70% of the W-EGFP cells were ciliated. Notably, the number of ciliated cells was dose-dependently decreased by siTmsb10 treatment (Fig. 4a, b), whereas siCnt treatment had no effect. The number of ciliated cells was decreased to 45% relative to baseline by KD with siTmsb10 at a concentration of 10.0 nM. Treatments with higher concentrations resulted in decreased survival of W-EGFP cells. Therefore, the following

**Fig. 3 Suppression of FLC differentiation by _Tmsb10_ KD. a** The experimental procedure for reconstruction of fetal testes. W-EGFP cells (E16.5) were plated and treated with siRNA for 24 h. Whole testicular cells prepared from wild-type fetuses (E16.5) were mixed with siRNA-treated W-EGFP cells to reconstruct testes. The detailed procedure for testis reconstruction and culture is described in the Materials and Methods. **b** A representative image of the reconstructed testis. Testicular tubules (tu) formed, and FLCs with strong EGFP staining were observed in the interstitial regions of the reconstructed testes. **c** The reconstructed testes were cultured in the presence (+) or absence (−) of SAG for 21 days. The reconstructed testes were examined under a fluorescence microscope to measure EGFP fluorescence. Scale bar = 100 μm. **d** The EGFP-positive cells (indicated by the relative EGFP-positive area) in the reconstructed testes above were analyzed quantitatively after incubation for 21 days. The cells were cultured in the absence (−, open bar) or presence of SAG (+, gray bar). $n = 3$. ***$p < 0.001$. **e** Representative images of the reconstructed testes using W-EGFP cells treated with the following siRNAs: control (si_Cnt_), _Tmsb10_ (si_T10_), _Tmsb4x_ (si_T4x_), and _Ad4BP/SF-1_ (si_Ad4_). Scale bar = 100 μm. **f** The EGFP-positive cells in the reconstructed testes above were analyzed quantitatively. Letters a, b, and c denote significant differences between the cell groups treated with si_Cnt_, si_T10_, and si_T4x_, and si_Ad4_. $n = 3$. $p < 0.001$. **g** Testes were reconstructed using W-EGFP cells treated with si_Cnt_ or si_T10_. They were cultured for 21 days in the presence of SAG. Scale bar = 100 μm. **h** The EGFP-positive cells in the reconstructed testes above were analyzed quantitatively. $n = 3$. ***$p < 0.001$. **i** W-EGFP cells were treated with si_Cnt_ (open bar) or si_T10_ (light blue bar). Expression of _Gli1_ in the W-EGFP cells was examined by qRT-PCR. The data were normalized by _Rn18s_ and are presented as means ± SEM. $n = 3$. **$p < 0.01$.

_Tmsb10_ KD experiments were performed using a concentration of 10.0 nM. A similar decrease in the number of ciliated cells was seen when W-EGFP cells cultured in the presence of SAG were subjected to _Tmsb10_ KD (Supplementary Fig. 6). By contrast, _Tmsb4x_ KD had no effect on ciliation (Supplementary Fig. 5d). Next, we examined whether ciliation and FLC differentiation were simultaneously suppressed by si_Tmsb10_ in a dose-dependent manner. Testes were reconstructed with W-EGFP cells treated with increasing concentrations of si_Tmsb10_. As shown in Fig. 4c, FLC differentiation was suppressed dose dependently by si_Tmsb10_. Likewise, si_Tmsb10_ dose-dependently decreased _Gli1_ gene expression in cultured W-EGDFP cells (Fig. 4d).

We next attempted to determine the mechanism whereby _Tmsb10_ regulates ciliogenesis. Regarding the function of TMSB10, experiments using human cancer cell lines showed that the protein binds directly to RAS, and thereby suppresses RAS-RAF interaction[32]. Moreover, this suppression ultimately resulted in failure of ERK phosphorylation and activation. Based on these findings, we planned to confirm the interaction between TMSB10 and RAS in W-EGFP cells with pull-down assays using ectopically expressed Flag-tagged TMSB10. Unfortunately, however, we could not find an efficient method for plasmid DNA transfection into W-EGFP cells. Thus, the interaction was confirmed in HEK293 cells (Fig. 4e). Subsequently, ERK phosphorylation, an event that occurs downstream of RAS/RAF activation, was examined in si_Cnt_- or si_Tmsb10_-treated W-EGFP cells. The amount of phosphorylated ERK (pERK) was increased by _Tmsb10_ KD in the absence of SAG (Fig. 4f, g). A few papers reported that hedgehog signal activates RAS/ERK pathway in several cell types[46,47]. To exclude the possibility that DHH signal activates RAS/ERK pathway in W-EGFP cells, we examined whether pERK is affected by SAG treatment. As the result, we found that the amount of pERK was not changed by the treatment, strongly suggesting that DHH signal does not affect RAS/ERK pathway in W-EGFP cells.

Taken together, these results indicated that TMSB10 promotes ciliogenesis and suppresses the RAS/ERK pathway. However, it remained unclear whether these two effects were connected. Several prior studies demonstrated that ciliogenesis was suppressed by the RAS/ERK pathway[48–50]. Therefore, as shown in Fig. 4h, we tentatively assumed that TMSB10 promotes ciliogenesis by suppressing the RAS/ERK pathway. To test this assumption, we examined the functional correlation between _Tmsb10_ and _Ras_ during FLC differentiation. W-EGFP cells treated with si_Tmsb10_ and/or si_Ras_ were used for testis reconstruction assays. Similar to the results shown in Fig. 3g, h, FLC differentiation was decreased by _Tmsb10_ KD (Fig. 4i). Consistent with the aforementioned assumption (Fig. 4h), FLC differentiation was increased by _Ras_ KD. Moreover, the decrease in FLC differentiation by _Tmsb10_ KD was canceled by simultaneous KD of _Ras_. The expression of the

_Gli1_ gene in cultured W-EGFP cells was similarly affected by treatment with these siRNAs (Supplementary Fig. 7).

We next examined the effect of _Ras_ KD on ciliogenesis. Again, _Tmsb10_ KD decreased the number of ciliated cells to 43% relative to control (Fig. 4j). Considering the effects of the KDs described above, we expected that _Ras_ KD would increase the number of ciliated cells. Unexpectedly, however, this was not the case. As noted above, at most ~70% of W-EGFP cells were ciliated even when they were cultured in serum-free medium, suggesting that this was the maximum percentage that could be ciliated under this condition. This assumption seemed to be supported by a double-KD study with si_Tmsb10_ and si_Ras_. Although KD of only _Ras_ had no effect, the decrease of ciliated cells by _Tmsb10_ KD was mitigated by _Ras_ KD, and their numbers reached those seen following treatment with si_Cnt_.

**PDGF possibly regulates ciliogenesis.** As indicated in Fig. 4h, the RAS/ERK pathway seemed to be involved in FLC differentiation. However, it remained unclear which molecule(s) activated RAS in W-EGFP cells. We considered PDGF to be a possible candidate because FLC differentiation was largely abrogated by _Pdgfra_ gene disruption[6], and PDGF signaling activated the RAS/ERK pathway[51].

First, we examined whether PDGF-AA activated FLC differentiation in reconstructed testes. As shown in Fig. 5a, FLC differentiation was not activated by PDGF-AA alone. Interestingly, however, PDGF-AA further enhanced the differentiation induced by SAG treatment. Likewise, _Gli1_ gene expression was increased by PDGF-AA in the presence but not the absence of SAG (Fig. 5b).

Next, we examined whether PDGF affected ERK phosphorylation in W-EGFP cells. Consistent with previous studies[51], PDGF-AA increased the amount of pERK, although the amount of ERK was unchanged (Fig. 5c, d). These results demonstrated that PDGF-AA increased pERK, probably by activating RAS. Since TMSB10 was shown to suppress the RAS/ERK pathway (Fig. 4f), it was expected that PDGF-promoted ERK phosphorylation would be increased by _Tmsb10_ KD, and this was found to be the case (Fig. 5e, f). Based on these observations, it is likely that PDGF, in the presence of HH signaling, promotes FLC differentiation by activating the RAS/ERK pathway (Fig. 5g).

However, we noticed the following inconsistency. As described above, the RAS/ERK pathway was shown to suppress ciliogenesis[48–50]. In addition, the present study demonstrated that PDGF activated the RAS/ERK pathway in W-EGFP cells. These observations suggest that PDGF activates the RAS/ERK pathway and then suppresses ciliogenesis, resulting in suppression of DHH-dependent FLC differentiation (Fig. 5g). Nevertheless, Fig. 5a shows that PDGF promotes FLC differentiation cooperatively with DHH.

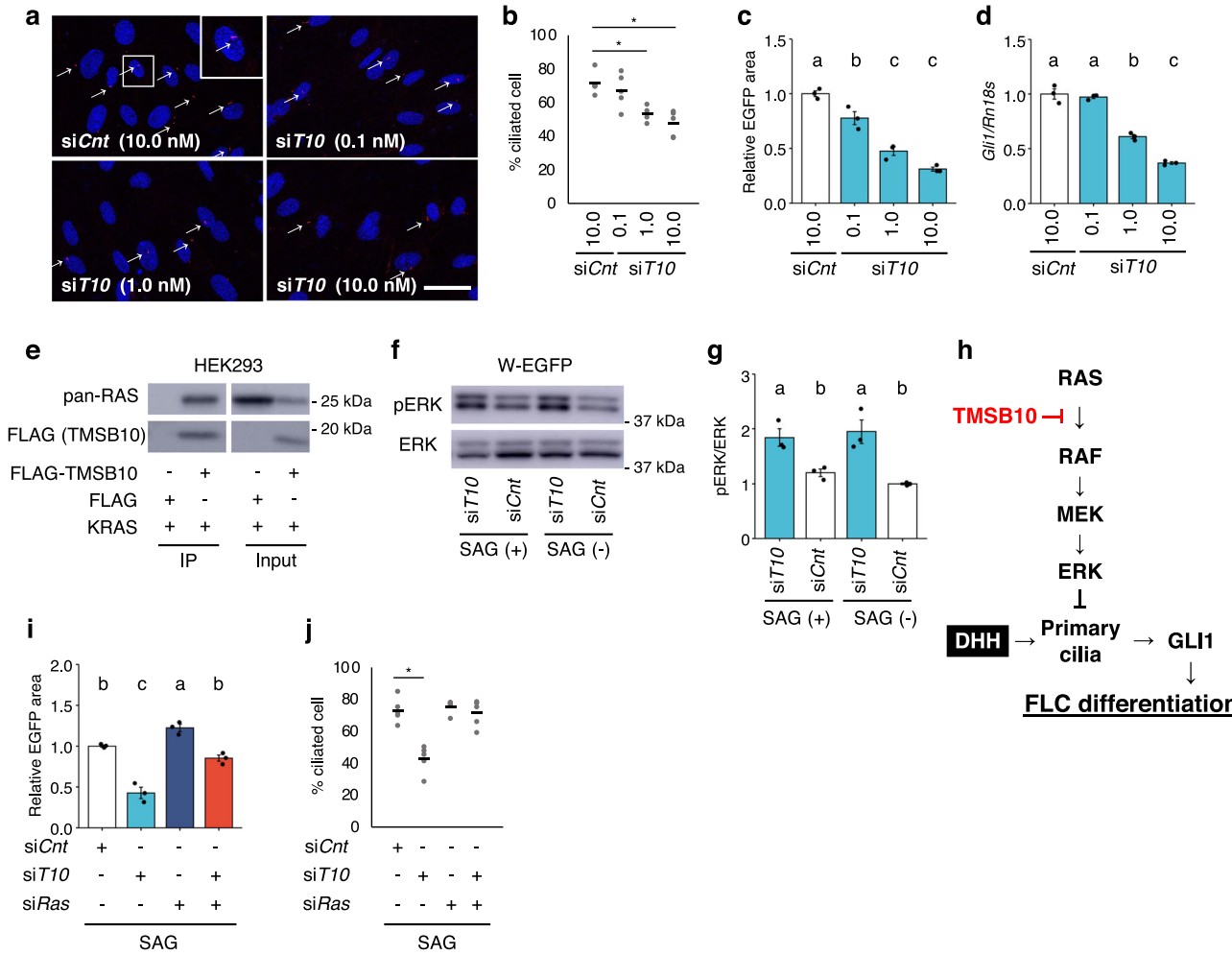

**Fig. 4 Role of *Tmsb10* in the regulation of ciliogenesis via suppression of the RAS/ERK pathway. a** E16.5 W-EGFP cells were treated with 0.1, 1.0, or 10.0 nM si*T10* or 10.0 nM si*Cnt*, then cultured in serum-free medium for 24 h. Thereafter, the cells were subjected to immunostaining for the cilial marker protein ARL13B (red). Nuclei were stained with 4′,6′-diamidino-2-phenylindole (DAPI) (blue). The enclosed area is enlarged at the top right. Arrows indicate primary cilia. Scale bar = 10 μm. **b** Ciliated cells detected in the studies above were counted. The ratios of the ciliated cells to all cells (%) are shown. $n = 5$. *$p < 0.001$. **c** W-EGFP cells were treated with 0.1, 1.0, or 10.0 nM si*T10* or 10.0 nM si*Cnt*, then subjected to testis reconstruction assay. The EGFP-positive cells (indicated by the relative EGFP-positive area) in the reconstructed testes were analyzed quantitatively after incubation for 21 days. $n = 3$. $p < 0.001$. **d** W-EGFP cells were cultured and treated with 0.1, 1.0, or 10.0 nM si*T10* or 10.0 nM si*Cnt*. Expression of the *Gli1* gene in the cells was examined. The data were standardized using *Rn18s*. $n = 3$. $p < 0.001$. **e** Interactions between TMSB10 and RAS were examined. Whole-cell lysates were prepared from HEK293 cells overexpressing FLAG-TMSB10 or FLAG together with KRAS. Proteins interacting with TMSB10 were immunoprecipitated with anti-FLAG antibodies. The immunoprecipitates were subjected to immunoblotting using antibodies for pan-RAS and FLAG. The positions of molecular weight markers are indicated on the left. Full blot images are displayed in Supplemental Fig. 9. **f** Whole-cell extracts were prepared from E16.5 W-EGFP cells treated with si*T10* or si*Cnt* in the presence (+) or absence (−) of SAG. Levels of phospho-ERK (pERK) and ERK were examined by western blotting. **g** Signal intensities in the blots above were quantified as described in the Materials and Methods. The amounts of pERK relative to ERK are shown. $n = 3$. $p < 0.01$. **h** A schematic illustration summarizing the results so far. TMSB10 was assumed to suppress the RAS/ERK pathway by interacting with RAS. **i** W-EGFP cells were treated with si*Cnt* (open bar), si*T10* (light blue bar), si*Ras* (dark blue bar), or both si*T10* and si*Ras* (red bar). These cells were then used for testis reconstruction. The reconstructed testes were cultured in the presence of SAG for 21 days. The EGFP-positive cells in the reconstructed testes were analyzed quantitatively. $n = 3$. $p < 0.001$. **j** W-EGFP cells were treated with siRNAs as above, and their effect on ciliogenesis was examined by immunostaining for ARL13B. Ciliated cells were counted and the ratios of these cells to all cells (%) are plotted. $n = 5$. *$p < 0.001$. Letters a, b, and c on the bars in **c**, **d**, **g**, and **i** denote significant differences between the cell groups.

This apparent inconsistency strongly suggests that another pathway functions downstream of the PDGF receptor to modulate the suppression of ciliogenesis by the RAS/ERK pathway. In fact, ciliogenesis did not seem to be markedly affected by PDGF-AA (Fig. 5h).

**Role of the PDGF-activated PI3K/AKT pathway in FLC differentiation.** It has been established that PDGF signaling activates the PI3K/AKT pathway, in which PIP₃, which is generated from PIP₂ by PI3K, activates AKT (Fig. 6a)[51,52]. Thus, we investigated the possibility that this pathway regulates FLC differentiation by activating ciliogenesis. First, we examined whether AKT phosphorylation was promoted by PDGF-AA in W-EGFP cells. As expected, the amount of phosphorylated AKT was increased in the presence of PDGF-AA, whereas that of AKT was not affected (Fig. 6b, c). Next, we examined whether the PI3K/AKT pathway regulates FLC differentiation in testis reconstruction assays using W-EGFP cells treated with siRNA for *Akt* and *phosphatase and*

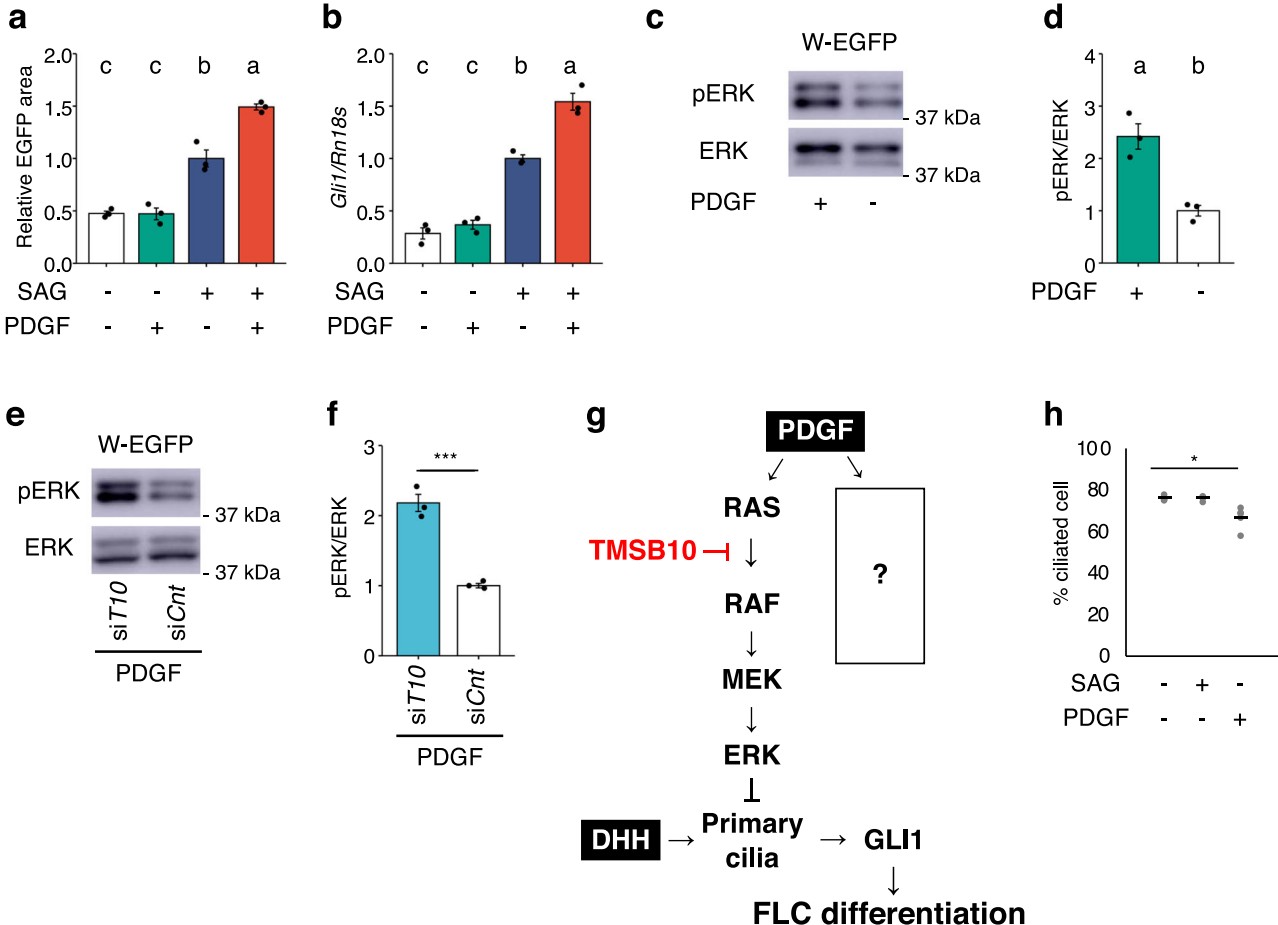

**Fig. 5 Activation of the RAS/ERK pathway by PDGF. a** Reconstructed testes were incubated in the absence (open bar) or presence of PDGF-AA (green bar), SAG (dark blue bar), or both PDGF-AA and SAG (red bar). The EGFP-positive cells in the reconstructed testes were analyzed quantitatively after incubation for 21 days. $n = 3$. $p < 0.001$. **b** W-EGFP cells were cultured under the same conditions as above. Expression of *Gli1* in the cells was examined by qRT-PCR. The data were normalized by *Rn18s* and are presented as means ± SEM. $n = 3$. $p < 0.01$. **c** Whole-cell extracts prepared from W-EGFP cells were cultured in the presence (+) or absence (−) of PDGF-AA and subjected to Western blotting for ERK and pERK. The position of a molecular weight marker is indicated on the left. **d** Signal intensities of the blots above were quantified as described in the Materials and Methods. The amounts of pERK relative to ERK are shown. Letters a, b, and c on the bars in a, b, and d denote significant differences between the cell groups. $n = 3$. $p < 0.01$. **e** W-EGFP cells were treated with siRNA for siT10 and thereafter cultured in the presence (+) or absence (−) of PDGF-AA. Whole-cell extracts prepared from the cells were subjected to western blotting for ERK and pERK. **f** Signal intensities of the blots above were quantified. The amounts of pERK relative to ERK are shown. $n = 3$. ***$p < 0.001$. Full blot images for **c** and **d** are displayed in Supplemental Fig. 9. **g** A tentative schema for the function of PDGF is shown. It was assumed that another signal pathway could be activated downstream of PDGF. **h** W-EGFP cells were cultured in the absence (−) or presence (+) of PDGF-AA and SAG. Ciliogenesis in the W-EGFP cells was examined by immunostaining for ARL13B. Ciliated cells were counted and the ratios of these cells to all cells (%) are plotted. $n = 3$. *$p < 0.001$.

*tensin homolog* (*Pten*). PTEN is known to suppress the PI3K/AKT pathway by mediating the conversion of PIP$_3$ to PIP$_2$ (Fig. 6a)[52]. When the reconstructed testes were incubated in the presence of SAG and PDGF-AA, *Akt* KD suppressed FLC differentiation. Conversely, *Pten* KD increased FLC differentiation (Fig. 6d). Likewise, the expression of *Gli1* in W-EGFP cells was decreased and increased by *Akt* KD and *Pten* KD, respectively (Fig. 6e). The *Tmsb10* KD-induced decreases in the FLC differentiation rate and in *Gli1* gene expression were observed in the presence of SAG ßand PDGF-AA.

We next examined the possible role of the PI3K/AKT pathway in the regulation of ciliogenesis. As shown in Fig. 6f, *Akt* KD decreased the number of ciliated W-EGFP cells in the presence of SAG and PDGF-AA. *Pten* KD was expected to increase the number of ciliated cells. However, as was the case with *Ras* KD (Fig. 4j), this increase was not observed.

Because several studies have demonstrated that activated AKT suppresses the RAS/ERK pathway by reducing RAF activity[53], we assumed that activating the PI3K/AKT pathway

would reduce ERK phosphorylation and eventually promote FLC differentiation through ciliogenesis. Therefore, we examined whether inhibition of the PI3K/AKT pathway impacted ERK phosphorylation. As expected, the PI3K inhibitor wortmannin decreased AKT phosphorylation and increased ERK phosphorylation in W-EGFP cells (Fig. 6g, h). By contrast, when PI3K/AKT signaling was activated by *Pten* KD, AKT phosphorylation and ERK phosphorylation were increased and decreased, respectively (Fig. 6i, j). Together, these results suggest that activation of the PI3K/AKT pathway promotes FLC differentiation by enhancing ciliogenesis via suppression of the RAS/ERK pathway (Fig. 6a, Supplementary Fig. 8).

## Discussion

FLCs are known to increase in number in the fetal testes until late pregnancy, even though they rarely proliferate. Therefore, this increase has been thought to be due to the differentiation of progenitor cells into FLCs[15–21]. The interstitial space of the developing

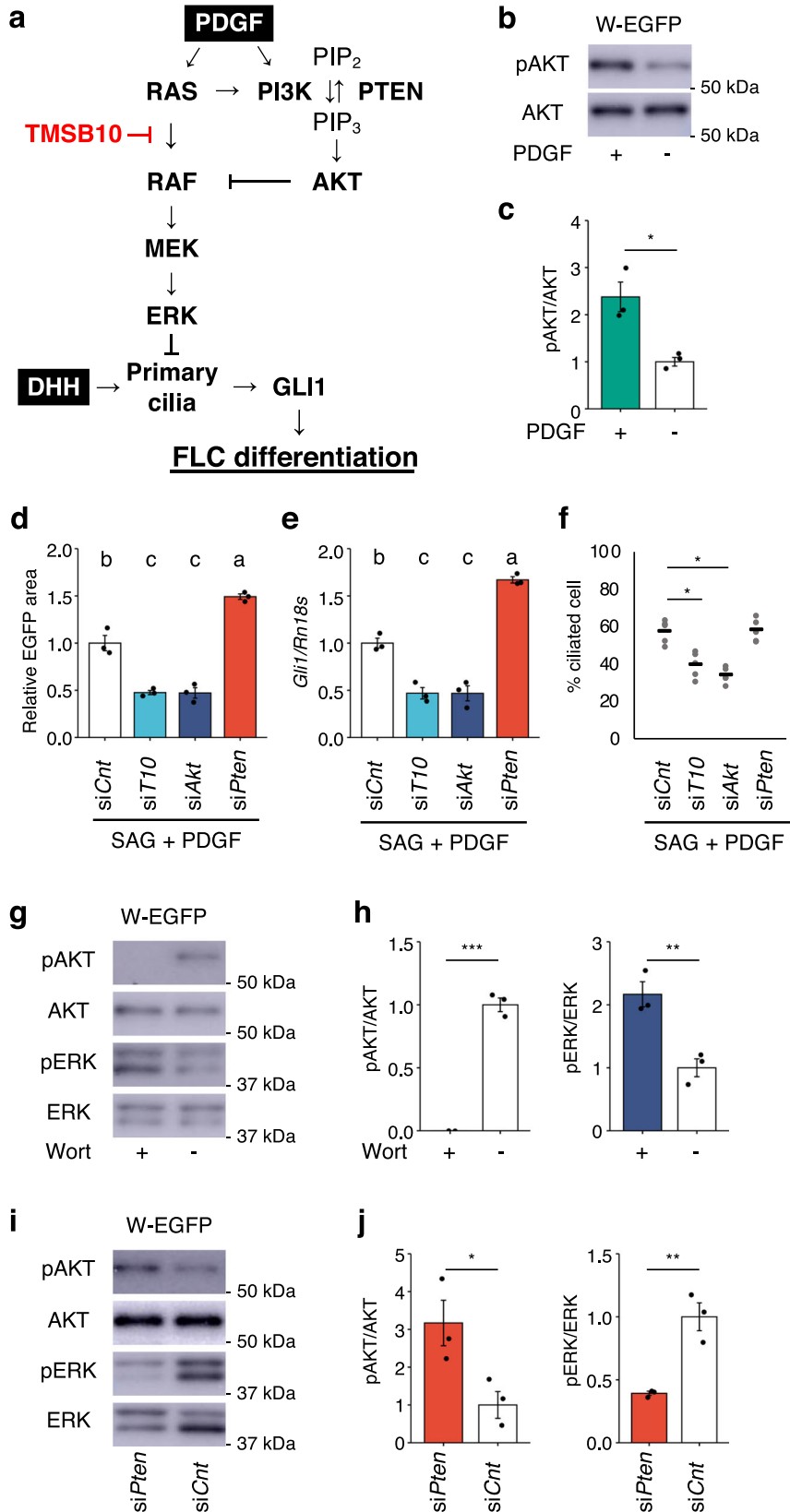

fetal testes is occupied predominantly by uncharacterized cells, in addition to smaller numbers of FLCs, peritubular myoid cells, endothelial cells, and macrophages. According to studies published so far, FLC progenitors are thought to be present in the interstitial space as cells positive for MAFB[16], ARX[9,15], and/or Notch[17,18] and Nestin[19]. In addition, our testis reconstruction study demonstrated

the presence of FLC progenitors among the uncharacterized interstitial cells (W-EGFP cells)[36]. Based on these findings, it seemed critical to determine what kinds of cells constitute the uncharacterized W-EGFP cell population.

scRNA-seq is a powerful technique for characterizing cells that contain one or more unidentified cell populations. Indeed, novel

**Fig. 6 Dual role of PDGF in regulating FLC differentiation. a** A schematic summary of signal pathways regulating FLC differentiation and in which *Tmsb10* acts as a suppressor. **b** W-EGFP cells were cultured in the presence (+) or absence (−) of PDGF-AA. Whole-cell extracts prepared from the cells were subjected to western blotting for phospho-AKT (pAKT) and AKT. Representative images of the blots are shown. The positions of molecular weight markers are indicated at the left. **c** Signal intensities in the blots above were quantified as described in the Materials and Methods. The amounts of pAKT relative to AKT are shown. $n = 3$. *$p < 0.05$. **d** W-EGFP cells were treated with siCnt, siT10, siAkt, or siPten, and then used for testis reconstruction. The reconstructed testes were cultured in the presence of SAG and PDGF for 21 days. The EGFP-positive cells in the reconstructed testes were quantified. $n = 3$. $p < 0.001$. **e** W-EGFP cells were cultured under the same conditions as above. Expression of *Gli1* in the cells was determined by qRT-PCR. The data were normalized by *Rn18s* and are presented as means ± SEM. $n = 3$. $p < 0.001$. Letters a, b, and c on the bars in **d** and **e** denote significant differences. **f** W-EGFP cells were cultured under the same conditions as above. Ciliogenesis in the W-EGFP cells was examined by immunostaining for ARL13B. $n = 5$. *$p < 0.001$. **g** W-EGFP cells were cultured in the presence (+) or absence (−) of wortmannin (Wort). Whole-cell extracts were subjected to western blotting for pAKT, AKT, pERK, and ERK. **h** Signal intensities in the blots above were quantified as described in the Materials and Methods. The amounts of pAKT relative to AKT (left) and pERK relative to ERK (right) are shown. $n = 3$. **$p < 0.01$. ***$p < 0.001$. **i** W-EGFP cells treated with siRNA for siPten and siCnt. Whole-cell extracts prepared from the cells were subjected to western blotting for pAKT, AKT, pERK, and ERK. Full blot images for **b**, **g**, and **i** are shown in Supplemental Fig. 9. **j** Signal intensities in the blots above were quantified. The amounts of pAKT relative to AKT (left) and pERK relative to ERK (right) are shown. $n = 3$. *$p < 0.05$. **$p < 0.01$.

---

cell populations have frequently been unveiled by this technique[54]. So far, studies have obtained single-cell transcriptomes from the cells constituting the fetal and postnatal testes[20,55–59]. These studies predicted the cellular lineages of FLCs and ALCs, as well as those of germ and Sertoli cells. Eventually, it was hypothesized that FLCs originate from uncharacterized interstitial cells. Unfortunately, however, none of the studies identified a particular cell population as segregating and differentiating into FLCs. Although we currently do not know why these studies did not identify the cells that we found in the present study, it might be because these single-cell studies utilized the overall population of testicular cells, which included germ cells. Therefore, their contents demonstrate high biological complexity, which might make it difficult to identify a population consisting of small number of differentiating cells.

In contrast to these studies, Stevant et al. utilized EGFP-positive somatic cells from *Nr5a1*-BAC-EGFP transgenic mice[20]. The developmental lineages of these cells were examined using several stages of the developing testes (E10.5 to E16.5). Eventually, the authors identified multipotent progenitor cells that gave rise to Sertoli and Leydig cells. Moreover, a recent study by the same group demonstrated that *Wnt5a*-expressing cells in E11.5 to E12.5 testes were potentially progenitors of both FLCs and ALCs[21].

Our current study utilized a simple cell population, namely W-EGFP cells, that probably corresponds to uncharacterized interstitial cells. Ultimately, we identified FLC progenitor cells characterized by the transient expression of *Tmsb10*. The utilization of a simple cell population may have emphasized subtle differences and thereby enabled us to identify the small population of FLC progenitors.

The developmental relation between progenitors characterized by *Wnt5a* versus *Tmsb10* might be interesting. Considering that the expression of *Wnt5a* reaches a peak at E12.5–E13.5 and declines at E16.5[21], it might be reasonable to assume that the multipotent progenitor cells of FLCs and ALCs are established at an early stage by the expression of *Wnt5a*. Thereafter, FLC progenitors may be defined more selectively by the expression of *Tmsb10*. Because the cells used in the two studies with *Wnt5a* and *Tmsb10* were derived during distinct developmental stages, the studies might successfully identify progenitor cells at distinct stages. Further investigations to determine the developmental relation between *Tmsb10*-positive and *Wnt5a*-positive cells, and also between previously identified *Mafb*-positive, *Arx*-positive, and *Notch*- and *Nestin*-positive cells might help us to clarify the entire process of FLCs differentiation.

It is known that both *Tmsb10* and *Tmsb4x* possess actin sequestration activity, and thereby suppress actin polymerization[29,30]. *TMSB10* was found to be widely expressed in a variety of developing tissues, such as tooth germ[60], antler growth center[61], post-implantation embryos[27], ovarian follicles[62], and brain[63]. In addition, elevated expressions of TMSB10 were observed in a variety of cancers[31]. Because overexpression of TMSB10 resulted in the disappearance of F-actin and thereby enhanced migration and invasion activities[32], the functions of TMSB10 in cancers have been investigated based on the assumption that the protein acts as an actin-mediated tumor suppressor.

In addition, an interesting study published in 2005 showed that TMSB10 hindered RAS-RAF interaction, thereby suppressing the RAS/ERK signaling pathway as well as angiogenesis and tumor growth[32]. Consistent with these findings, another study indicated that TMSB10 reduced cancer cell activities by suppressing the ERK pathway[31]. Together, these studies indicated that decreased expression of TMSB10 promotes tumor growth. In contrast, however, increased expression of TMSB10 was associated with high malignant potential in cancers[31]. Interestingly, a study of breast cancer revealed that enhanced expression of TMSB10 promoted the proliferation and migration of cancer cells by activating AKT/FOXO signaling[64]. Although the functions of TMSB10 in cancer cells remain controversial, these findings regarding the potential of TMSB10 to regulate signal pathways showed that it has functions beyond those related to actin sequestration. In fact, our current study demonstrated that TMSB10 suppresses the RAS/ERK pathway in W-EGFP cells. Moreover, we found that *Tmsb10* promotes the differentiation of FLC progenitors.

PDGF plays crucial role in the differentiation of various cell types by activating the RAS/ERK and PI3K/AKT pathways[51]. As for the functional correlation between the two pathways, it was found that the PI3K/AKT pathway may suppress the RAS/ERK pathway, and thus these two pathways exert opposing effects on cell differentiation[53]. Likewise, our present study demonstrated that PDGF can activate both the RAS/ERK and PI3K/AKT pathways, which suppress and promote FLC differentiation, respectively. This bidirectional activity of PDGF may maintain the balance between promotion and suppression of FLC differentiation, and thereby sustain the ability of some W-EGFP cells, if not all, to differentiate into FLCs.

It has been established that both DHH and PDGF are required for FLC differentiation[5,6,13,14]. However, it remains unclear whether these growth factors function cooperatively or independently in the differentiation process. Regarding this issue, our testis reconstruction assays, fortunately, showed that PDGF promotes FLC differentiation only in the presence of HH signaling. This finding strongly suggested the presence of crosstalk between PDGF and DHH signals. Eventually, we found that PDGF regulates the formation of cilia required for DHH signal transduction, confirming that crosstalk occurs between the two growth factors.

A previous study demonstrated crosstalk between PDGF and TGFβ during osteogenic differentiation[65]. Interestingly, the study showed that TGFβ-induced osteogenic differentiation was markedly enhanced by PDGF even though PDGF alone failed to promote the differentiation. Another study revealed that lens differentiation was regulated by antagonistic interaction between the PDGF-driven RAS/ERK pathway and the FGF-driven PI3K/AKT pathway[66]. In addition to the crosstalk between PDGF and both TGFβ and FGF, we revealed the presence of crosstalk between PDGF and HH signaling. Moreover, it has been known that NOTCH is involved in FLC differentiation[7,19] and the molecule cross-talks with Ras pathway[67]. These findings suggest that NOTCH signaling may also be involved in the regulation of ciliogenesis.

Our current study investigated *Tmsb10*, which was identified by single-cell transcriptome analyses of the uncharacterized interstitial cell population of the fetal testes, and provided several fundamental clues regarding the mechanism underlying FLC differentiation. To obtain a more comprehensive understanding of FLC differentiation, it might be critical to identify the mechanism whereby *Tmsb10* expression is selectively induced in a certain population of interstitial cells and how it is transiently expressed prior to differentiation into FLCs.

## Methods

### Preparation of EGFP cells with strong (S-EGFP) or weak (W-EGFP) staining.
A DNA fragment, FLE (fetal Leydig enhancer), which induces FLC-specific gene expression was isolated from the *Ad4BP/SF-1* gene. By using the fragment, a transgene, *FLE-EGFP*, was constructed and then used to establish *FLE-EGFP* transgenic mice in which FLCs are labeled by EGFP[33]. EGFP-labeled transgenic testes were harvested at E16.5, and were dispersed with collagenase (0.1 U/ml, Sigma-Aldrich, St. Louis, MO, USA) and dispase (1 U/ml, Thermo Fisher Scientific, Waltham, MA, USA), at 37 °C for 30 min[34]. After pipetting gently, the gonads were incubated with DNase I (0.2 mg/ml, Roche Diagnostics Corp., Indianapolis, IN, USA) at 37 °C for 15 min. Using JSAN (Bay bioscience, Kobe, Japan), the dispersed cells were fractionated by FACS into N-EGFP (negative for EGFP), W-EGFP (weakly positive for EGFP), and S-EGFP (strongly positive for EGFP) populations based on the negative, weak, or strong EGFP fluorescence intensity, respectively. All protocols for the animal experiments were approved by the Animal Care and Use Committee of Kyushu University, and all experiments were performed in accordance with the institutional guidelines.

### scRNA-seq.
Single S-EGFP or W-EGFP cells were plated by FACS (SH800, Sony, Tokyo, Japan) into individual wells of a 384-plate (Piko PCR Plate, Thermo Fisher Scientific) pre-loaded with lysis buffer. The CEL-Seq2 protocol established by Hashimshony et al.[37] was used for RNA extraction and library preparation. Briefly, the RNA of each cell was reverse transcribed using CEL-Seq primers containing an anchored poly(T), a 6-bp unique molecular identifier, a 5' Illumina adapter (San Diego, CA, USA), a cell-specific 6-bp barcode, and a T7 promoter (Supplementary Table 2). The External RNA Controls Consortium spike-ins (Thermo Fisher Scientific) were added to each preparation. After second-strand synthesis reaction, the double-stranded cDNAs were transcribed in vitro by T7 RNA polymerase. The synthesized RNAs were reverse transcribed using random primers with the 3' Illumina adapter. Finally, the libraries were amplified by PCR (11 cycles). The pair-ended CEL-Seq2 libraries were sequenced by HiSeq 2500 (Illumina).

### Data analysis of single-cell transcriptomes.
The quality check of the raw sequence reads was performed using FastQC (version 0.11.7), and thereafter the reads were analyzed according to the CEL-Seq2 pipeline[37]. First, the reads of CEL-Seq2 libraries were demultiplexed to each cell using the CEL-Seq barcodes. To identify the transcript, the reads were mapped to the mouse reference genome (mm10) by Bowtie 2 software (version 2.3.4.1)[68]. PCR duplicates were removed using UMI information. The mapped reads were counted using HTSeq (version 0.9.1)[69]. The quality of the sequence data was further evaluated; low-quality samples were removed by setting certain thresholds of low total reads, few expressed genes (<1000 genes), and high spike-in proportions[70].

Cells in the G1 cell cycle phase were selected to avoid potential confounding effects from cell cycle-induced differences. Expression levels of the remaining genes were normalized and denoised. These data were subjected to hierarchical clustering on the principal components to divide cells into clusters. Differentially expressed genes (DEGs) of the clusters were identified to characterize the clusters. R packages including SingleCellExperiment (version 1.0.0), scater (version 1.6.3), scran (version 1.6.9), Seurat (version 3.0.2), and Monocle (version 2.6.4) were used.

### siRNA treatments.
W-EGFP cells were cultured on Advanced TC 24-well plates (Greiner Bio-One, Kremsmünster, Austria) in α-modified Eagle's medium (α-MEM, Nacalai Tesque, Kyoto, Japan) supplemented with 10% fetal bovine serum (FBS, Thermo Fisher Scientific) and penicillin and streptomycin (PS, Thermo Fisher Scientific) at 37 °C under 5% $CO_2$ for 24 h. The cells were transfected with siRNAs using lipofectamine RNAiMAX reagent (Thermo Fisher Scientific) for 24 h. The siRNAs used in this study are listed in Supplementary Table 3. A control siRNA (Stealth RNAi Negative Control Medium GC Duplex; Thermo Fisher Scientific) was used as a negative control. The siRNA-treated cells were then utilized for qRT-PCR analyses, in vitro testis reconstruction, immunocytochemistry, and Western blotting. To investigate the effects of growth factor signals, the siRNA-treated cells were further cultured in the presence of SAG (0.5 μM; Adipogen Life Sciences, San Diego, CA, USA), mouse PDGF-AA (10 ng/mL; Sigma-Aldrich), wortmannin (0.1 μM; Cayman, Ann Arbor, MI, USA), or dimethyl sulfoxide (DMSO, Nacalai Tesque) as a negative control.

### Immunostaining.
Cryosections (10 μm) of mouse fetal testes and reconstructed testes attached to slide glasses were boiled for 5 min in 10 mM sodium citrate (pH 6.0) to unmask antigen epitopes[36]. The sections were incubated with primary antibodies in blocking buffer (2% skim milk (WAKO, Tokyo, Japan) in PBS) overnight at 4 °C, and subsequently with secondary antibodies in the blocking buffer for 1 h at room temperature. The primary and secondary antibodies used in this study are listed in Supplementary Table 4. Nuclei were counterstained with DAPI (Sigma-Aldrich).

To investigate ciliogenesis by immunofluorescence, W-EGFP cells cultured on μ-Plate 96-well TC ($2.0 \times 10^4$ cells/well) (ibiTreat, ibidi, Martinsried, Germany) in the same medium as above for 24 h were treated with the siRNAs described above. After the siRNA treatment, the cells were cultured in this medium again for 24 h. Thereafter, they were fixed with 4% paraformaldehyde in PBS for 15 min at room temperature, and permeabilized with 0.5% Triton X-100 (WAKO) in PBS for 10 min followed by incubation in blocking buffer (1% skim milk in PBS) for 20 min at room temperature. Subsequently, the cells were treated with mouse anti-ARL13B antibody overnight at 4 °C, and then with Alexa Fluor 555-labeled goat anti-mouse IgG antibody for 1 h at room temperature (Supplementary Table 4). Nuclei were counterstained with DAPI.

Immunofluorescence images were captured using an LSM 700 confocal laser scanning microscope (Carl Zeiss, Jena, Germany) and a BZ-X700 fluorescence microscope (Keyence, Osaka, Japan). The effects of the siRNA, SAG, and PDGF treatments on ciliogenesis were evaluated as the percentage of ciliated cells in at least 500 overall cells in each sample ($n = 5$).

### qRT-PCR analyses.
Total RNAs (50 ng) prepared using the RNeasy Micro Kit (Qiagen, Hilden, Germany) were reverse-transcribed to cDNA using Moloney Murine Leukemia Virus reverse transcriptase (Thermo Fisher Scientific) and random hexamers (Sigma-Aldrich)[71]. qRT-PCR was performed with a CFX96 Real-Time PCR Detection System (Bio-Rad Laboratories, Hercules, CA, USA) using SYBR Select Master Mix (Thermo Fisher Scientific). Gene expression was determined using the standard curve method. Gene expression levels were normalized to those of *Rn18s* (18 S ribosomal RNA). The primers used for qRT-PCR are listed in Supplementary Table 5.

### In vitro testis reconstruction.
For fetal testis reconstruction[36,72], wild-type testes at E16.5 were incubated in 0.25% trypsin/PBS (Sigma-Aldrich) at 37 °C for 10 min. The whole testicular cells ($8.0 \times 10^5$ cells) were mixed with W-EGFP cells ($2.0 \times 10^4$ cells) prepared from fetal testes at E16.5 or W-EGFP cells treated with siRNA ($2.5 \times 10^4$ cells). The reconstructed testes were cultured on a V-shaped agarose gel for 2 days and then transferred onto a bowl-shaped agarose gel, followed by culturing in α-MEM containing 10% Knockout Serum Replacement (Thermo Fisher Scientific) and PS at 37 °C under 5% $CO_2$. To investigate the effect of HH and PDGF signals on FLC differentiation, the reconstructed tissues were treated with SAG (0.5 μM), mouse PDGF-AA (10 ng/mL), or DMSO. The reconstructed tissues were observed under a BZ-X700 fluorescence microscope (Keyence) to capture EGFP-fluorescent and bright-field images. The differentiation rate was calculated based on the EGFP-positive area and the bright-field area[73].

### Western blotting analysis.
Whole-cell extracts were prepared from W-EGFP cells using lysis buffer (50 mM Tris–HCl (pH 8.0), 50 mM NaCl, 1 mM EDTA, and 1% SDS) containing phosphatase inhibitors (PhosSTOP tablet; Roche Diagnostics Corp.). After the protein concentration was determined using a BCA Protein Assay Kit (Pierce Biotechnology, Rockford, IL, USA), 5 μg of the whole-cell extract was subjected to SDS–polyacrylamide gel electrophoresis, followed by electrophoretic transfer to polyvinylidene fluoride membranes (Thermo Fisher Scientific). The membranes were incubated for 30 min at room temperature in Blocking One (Nacalai Tesque). The membranes were treated with primary antibodies in a reaction buffer (10% Blocking One in Tris-buffered saline; 10 mM Tris–HCl (pH 7.4), 150 mM NaCl, 0.05% Tween 20) overnight at 4 °C, and thereafter with horseradish peroxidase-conjugated secondary antibodies in the reaction buffer for 1 h at room temperature. The primary and secondary antibodies used in this study are listed in Supplementary Table 6. Washed membranes were developed using Chemi-Lumi One (Nacalai Tesque), and the images were captured using a lumino image analyzer (ImageQuant LAS 500, GE Healthcare, Buckinghamshire, UK). All images were quantified using Image Lab 6.0.0 software (Bio-Rad).

**Preparation of expression plasmids and a donor plasmid**. Full-length cDNAs of *Tmsb10* and *Kras* was amplified by PCR with sets of primers (Supplementary Table 7), and were used to construct p3xFLAG-TMSB10 and pCMV-HA-KRAS expression plasmids, respectively. The p3xFLAG-CMV10 expression plasmid (Sigma-Aldrich) was used as a control study.

To construct a donor plasmid for CRISPR/Cas9 technology, an 849-bp fragment upstream from the first ATG and an 849-bp fragment downstream from the first ATG of the *Tmsb10* gene were amplified from the C57BL/6 genome. These fragments were used as the 5' and 3' homologous arms. *mCherry* tagged with human influenza hemagglutinin (HA) and Thoseaasigna virus 2 A (T2A) at the N- and C-terminal sites (HA-mCherry-T2A), respectively, was synthesized as follows. A DNA fragment encoding *mCherry* was amplified from pmCherry-N1 (Clontech, Palo Alto, CA, USA) and inserted into the EcoRI/BglII site of pCMV-HA (Takara, Shiga, Japan) to generate pCMV-HA-mCherry. Thereafter, the plasmid was subjected to PCR with a 3' primer containing the T2A sequence to generate the HA-mCherry-T2A fragment. The primers used for amplification of the fragments are listed in Supplementary Table 7. Then, these three fragments (5' homologous arm, HA-mCherry-T2A, and 3' homologous arm) were inserted into the SalI/EcoRI site of the pBluescript II KS+ using an In-Fusion HD cloning kit (Clontech) (Supplementary Fig. 2).

**Physical interaction between TMSB10 and KRAS**. p3xFLAG-TMSB10, p3xFLAG-CMV10, and pCMV-HA-KRAS were transfected using the lipofectamine 2000 reagent (Thermo Fisher Scientific) in HEK293 cells. The cells were lysed in 50 mM Tris–HCl (pH 8.0), 300 mM NaCl, 10% glycerol, 1.5 mM MgCl$_2$, 1 mM EDTA, and 1% Triton X-100. The p3xFLAG-TMSB10 and p3xFLAG-CMV10 were immunoprecipitated with anti-FLAG antibody-conjugated magnetic beads (Sigma-Aldrich), and then the beads were sequentially washed three times with washing buffer (20 mM HEPES-KOH (pH 7.6), 100 mM KCl, 10% glycerol,1 mM EDTA, and 0.05% Tween20), and once with PBS. Finally, the immunoprecipitates were eluted from the beads with 10 μl elution buffer (50 mM Tris–HCl (pH 7.4), 150 mM NaCl, 500 μg/ml FLAG peptide). Eluates and inputs were subjected to SDS–polyacrylamide gel electrophoresis followed by Western blotting. The antibodies used in this study are listed in Supplementary Table 6.

**Generation of Tmsb10-mCherry knock-in mice**. A mouse line carrying the *mCherry* reporter gene at the *Tmsb10* locus (*Tmsb10-mCherry* KI) was generated using CRISPR/Cas9 technology[74,75]. A guide RNA (gRNA) was designed to target the transcription start site of *Tmsb10* using CRISPRdirect (http://crispr.dbcls.jp/). The gRNA was synthesized and purified using a CUGA7 gRNA Synthesis Kit (Nippon Gene, Tokyo, Japan). Oocytes were collected from F1 hybrid (C57BL/6 × DBA/2) BDF1 female mice that were superovulated by standard procedures and fertilized in vitro with sperms from male mice of the same genetic background. CAS9 protein (100 ng/μl; Nippon Gene), the gRNA (250 ng/μl each), and the donor plasmid were microinjected into the cytoplasm on one side of the blastomere at the two-cell stage. The cells were transferred to pseudo-pregnant ICR female mice. Genotypes of the pups were analyzed by PCR. Nucleotide sequences of the primers for gRNA preparation and genotyping are shown in Supplementary Table 7. After backcrossing more than five times with C57BL/6 J, *Tmsb10-mCherry* knock-in mice were further crossed with *FLE-EGFP* mice to generate *FLE-EGFP;Tmsb10-mCherry* mice. *FLE-EGFP* and *FLE-EGFP;Tmsb10-mCherry* male mice were crossed with ICR females (Japan SLC, Shizuoka, Japan). All protocols for the animal experiments were approved by the Animal Care and Use Committee of Kyushu University and the Animal Care and Use Committee of the National Research Institute for Child Health and Development. All experiments were conducted in accordance with institutional guidelines.

**Statistics and reproducibility**. At least three biologically independent samples were used in all experiments. All data are presented as mean ± standard error of the mean. We used Student's *t* tests for comparisons between two groups, and one-way analysis of variance followed by Tukey's multiple comparison test for multiple group comparisons. $p < 0.05$ was considered to indicate statistically significant differences between groups. The statistical analyses were performed using RStudio (Version 1.1.453 and 1.2.5033) with R software version 3.4.3 and 3.6.3 (https://www.r-project.org) and Microsoft Excel (Version 16.59).

**Reporting summary**. Further information on research design is available in the Nature Research Reporting Summary linked to this article.

## Data availability

scRNA-seq data have been deposited in the DNA Data Bank of Japan under the accession code DRA013467 (http://trace.ddbj.nig.ac.jp/DRASearch/). The plasmids of p3xFLAG-TMSB10 (#191133), pCMV-HA-KRAS (#191134), pCMV-HA-mCherry (#191136), and pKnockinDonor-HA-mCherry-T2A-Tmsb10 (#191137) are available on Addgene. The unedited blot images are included in Supplementary Fig. 9. All source data used for generating graphs in the main figures are found in Supplementary Data 1. The data that support the findings in this study are available from the corresponding author upon reasonable request.

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

## Acknowledgements

We thank Dr. Junko Sasaki (Medical Research Institute, Tokyo Medical and Dental University) for her technical advice and discussion. We are profoundly thankful to Prof. Chia-Yih Wang (Department of Cell Biology and Anatomy, College of Medicine, National Cheng Kung University) for his advice and discussion about primary cilia analysis. We appreciate the technical assistance of The Research Support Center, Research Center for Human Disease Modeling, Kyushu University Graduate School of Medical Sciences. This work was supported by JSPS KAKENHI Grant Numbers JP20K08863 (T.B.), JP17H06427 (T.B., K.-I.M.), JP20H03436 (K.-I.M.), JP20H04935 (S.T.), JP19H05244 (Y.O.), JP20H00456 (Y.O.), JP20H04846 (Y.O.), and JP20K21398 (Y.O.); by JST CREST Grant Number JPMJCR16G1 (Y.O.); and by AMED under Grant Numbers JP20gk0210019 (K.-I.M.) and JP20ek0109489h0001 (Y.O.).

## Author contributions

M.I., T.B., Y.S., and K.-I.M. conceived, designed, and conducted the experiments, and performed data analyses. F.T. and Y.O. constructed the scRNA-seq libraries and obtained transcriptomes. M.I., S.Y., D.S., and M.S. analyzed the sequence data. M.T. and S.T. produced genome-edited mice. K.S. and T.M. analyzed cell ciliation. M.I. and K.-I.M. prepared the manuscript.

## Competing interests

The authors declare no competing interests.
