## [Peer Review File · Communications Biology]

Reviewers' comments:

Reviewer #1 (Remarks to the Author):

Communications Biology

COMMENTS TO THE EDITORS AND THE AUTHORS

Manuscript ID COMMSBIO-22-0917-T: "Suppression of the RAS/ERK pathway by Tmsb10 is the key step for promoting fetal Leydig cell differentiation from progenitors"

BRIEF SUMMARY OF THE MANUSCRIPT

The authors stated that the study is aimed to investigate the mechanisms regulating the role of the growth factors in fetal Leydig cell (FLC) differentiation. The single-cell RNA sequencing was used to characterizing FLC progenitor cells. The results suggested that thymosin b10 (Tmsb10) was transiently upregulated in the progenitors. Besides, results suggested that platelet-derived growth factor (PDGF) activation of the RAS/ERK and PI3K/AKT signaling regulated ciliogenesis and promoted desert hedgehog (DHH)-dependent FLC differentiation. Tmsb10 expressed in the progenitor cells induced their differentiation into FLCs by suppressing the RAS/ERK pathway. The authors concluded that this study unveiled the molecular process of FLC differentiation and showed that it is cooperatively induced by DHH and PDGF.

OVERALL IMPRESSION OF THE WORK

The manuscript provides novel information related to fetal Leydig cells differentiation. The authors precisely applied several state-of-the-art approaches to prove the conclusions. The methodology is described very clearly providing enough evidence to repeat the methods. The statistical analyses are well applied. The figures are very clearly organized and very easy to follow even without legends. The images of the blots and images showing the results from the immunofluorescence studies are very neat. The supplemental tables and figures are extremely helpful. Although text describes the complex approach and the results, the text of the manuscript is easy to follow. The conclusions are supported by the results.

THE MINOR COMMENTS

Page 4, Lines 40-41: Please consider rewriting the sentence "...Sertoli cells provide germ cells with nutrients and stimuli to support their differentiation...". Probably the authors wanted to keep text simple, but the function of Sertoli cells is not only to "provide germ cells...". Sertoli cells secrete wide-variety of regulatory molecules including inhibins, activins, MIH, androgen binding protein etc. All of them are very important for regulation of Leydig cells and testicular homeostasis.

Page 4, Lines 44-45: Please consider rewriting the sentence "Unlike other vertebrates, mammalian species possess two developmentally different Leydig cells..." considering the recent advances in the field discussing the similar origin of the fetal and adult Leydig cells. Please cite original recent articles instead of review paper from 2018.

I would greatly appreciate if you will contact me if you find something in my comments is unclear/incorrect.

Good luck and all the best!

Reviewer #2 (Remarks to the Author):

In this manuscript, Inoue et al have used a mouse model where interstitial cells of the fetal testis are labelled to isolate sub-populations of fetal Leydig cells (FLC) that weakly (W-EGFP) or strongly (S-EGFP) express GFP under the control of a modified SF1/Ad4BP promoter. The S-EGFP correspond to fully differentiated FLCs. Single-cell RNA-Seq and testis reconstruction assays revealed that the W-EGFP cells are likely precursor, or undifferentiated, FLCs. The W-EGFP cells transiently express the Tmsb10 gene which was found to promote the differentiation of the precursor cells (W-EGFP) into FLCs (S-EGFP) by repressing the RAS/ERK pathway. The authors then went on to show that the DHH and PDGF pathways, both known to induce the differentiation of stem Leydig cells into FLCs, work together in a common pathway (cilia formation). This constitutes an important step forward as the mechanism of action of DHH and PDGF in Leydig cell differentiation remained unknown.

The authors have used elegant and powerful experimental designs. All experiments were properly planned and executed with the right controls. The results are convincing.

This reviewer believes that the manuscript could be improved and invites the authors to consider the following points:

1. Throughout the manuscript, the authors use the word "progenitors" to describe FLCs that are not differentiated. However, progenitor is a generic term and often leads to confusion. Is it used here to indicate stem Leydig cells of the fetal population? Or FLCs that have begun, but not completed, their differentiation process (stem, progenitor, partly differentiated)?
2. The authors conclude that Tmsb10 is key for Leydig cell differentiation, as mentioned in the title of the manuscript. Although a strong possibility, this conclusion is entirely based on in vitro differentiation/reconstruction assays. Another experimental approach, such as knocking out the Tmsb10 gene, would strengthen this conclusion. The authors have used gene editing to knock-in a mCherry cDNA in frame with the Tmsb10 gene in order to determine where/when Tmsb10 is expressed. With the gene editing approach established (guides, homology arms, constructs), this approach could be easily adapted to knockout the Tmsb10 gene.
3. Lines 99-107: it seems that this section would be more appropriate for the Introduction section. It is all previously published information.
4. Lines 101-102: S-EGFP are defined as FLCs. However, W-EGFP are identified as interstitial cells. The interstitium contains several cell types, including stem FLCs and stem ALCs. Could the large number of W-EGFP cells correspond to various interstitial cell types and not only "precursor" FLCs?
5. Since some Tmsb10 positive cells are endothelial cells (co-localize with laminin), is it possible that the W-EGFP could represent late differentiating FLCs from endothelial origin while the S-EGFP would have a different origin and have differentiated a bit earlier? FLCs are known to have different origins.
6. Do W-EGFP cells express classical steroidogenic markers (of FLCs and/or ALCs) when they become S-EGFP in the in vitro reconstruction/differentiation system?
7. The authors integrate the PDGF signalling pathway and showed that PDGF-AA increases pERK probably via the RAS pathway. Does PDGF-AA have any effect on Tmsb10 expression?
8. To this reviewer, there are other possibilities to explain the crosstalk between the PDGF, DHH, and possibly the NOTCH pathway (which is not really considered by the authors). NOTCH is known to keep Leydig cells in an undifferentiated state. In addition, there is evidence that the NOTCH and RAS pathways often intersect, either in a synergistic or antagonistic manner. Have the authors considered that NOTCH could work with the RAS pathway to increase pERK maintaining FLCs in an undifferentiated state? At least, the various options (involvement of NOTCH, role of PDGF on Tmsb10

expression, alternate pathways) should be discussed.

Reviewer #3 (Remarks to the Author):

In the manuscript titled "Suppression of the RAS/ERK pathway by Tmsb10 1 is the key step for promoting fetal Leydig cell differentiation from progenitors" authors investigated the molecular process of fetal Leydig cell differentiation and claimed to have demonstrated that FLC differentiation it is cooperatively induced by DHH and PDGF. In general, there are several limitations in the manuscript, some of which are as follows-

In Figure 3C-G, images are blurry and appears to have different magnifications making it hard to draw any conclusions.

Fig 4A, the immunostaining results appears to be nonspecific and this non specificity is reflected in Fig 4D where, siT10 dose incremental treatment induced non-significant changes in the DHH signaling.

In Fig 4H authors depicted an illustration of potential mechanisms of FLC differentiation. However, the results in Fig 4J contradicts the proposed mechanism as inhibiting RAS should not increase the expression of GLI.

It is unclear why in Fig 4F, authors compared the impact of presence or absence of SAG on pERK in context of siT10 when phosphorylation of ERK according to the illustration in Fig 4h is upstream of DHH signaling?

Similarly, It is unclear why in Fig 5C, authors compared the impact of presence or absence of SAG on pERK in context of PDGF when phosphorylation of ERK according to the illustration in Fig 4H and 5G is upstream of DHH signaling?

Importantly, authors should perform gain and loss of function experiments to validate the proposed mechanism in illustrations 4H, 5G and 6A in context of each of the candidates included.

Dear Reviewers,

We are submitting our revised manuscript '**Suppression of the RAS/ERK pathway by Tmsb10 is the key step for promoting fetal Leydig cell differentiation from progenitors**'. We greatly appreciate the valuable comments including suggestions and discussions from the reviewers. According to the comments, we have performed additional experiments and revised original figures and text. Our point-by-point responses to the reviewers' comments are shown below. The revised parts in the text were highlighted in the revised manuscript.

We thank the reviewers again for reviewing our manuscript, and we hope that we have addressed the reviewer's concerns appropriately.

Reviewer #1: 1. Page 4, Lines 40-41: Please consider rewriting the sentence "...Sertoli cells provide germ cells with nutrients and stimuli to support their differentiation...". Probably the authors wanted to keep text simple, but the function of Sertoli cells is not only to "provide germ cells...". Sertoli cells secrete wide-variety of regulatory molecules including inhibins, activins, MIH, androgen binding protein etc. All of them are very important for regulation of Leydig cells and testicular homeostasis.

According to the suggestion by the reviewer, we revised the sentence (Page 4, Line 42-45 in the revised manuscript). The original and revised sentences are below.

Original; Sertoli cells provide germ cells with nutrients and stimuli to support their differentiation, while Leydig cells produce a potent androgen, testosterone, that regulates the differentiation and functions of both germ and Sertoli cells¹.

Revised; Sertoli cells provide germ and Leydig cells with nutrients and regulatory molecules to regulate their differentiation and functions, and in turn, Leydig cells produce a potent androgen, testosterone, that regulates the differentiation and functions of both germ and Sertoli cells¹.

reference

1 Svingen, T. & Koopman, P. Building the mammalian testis: origins,

differentiation, and assembly of the component cell populations. *Genes & development* **27**, 2409-2426, doi:10.1101/gad.228080.113 (2013).

2. Page 4, Lines 44-45: Please consider rewriting the sentence “Unlike other vertebrates, mammalian species possess two developmentally different Leydig cells...” considering the recent advances in the field discussing the similar origin of the fetal and adult Leydig cells. Please cite original recent articles instead of review paper from 2018.

Considering the comment by the reviewer, we thought that ‘developmentally different’ may be inappropriate. Therefore, we decided to delete the words ‘developmentally different’ and the sentence was changed as follows (Page 4, Line 46-49 in the revised manuscript).

Original; Unlike other vertebrates, mammalian species possess two developmentally different Leydig cells, fetal-type Leydig cells (FLCs) and adult-type Leydig cells (ALCs), the former of which play pivotal roles in the masculinization of fetuses through androgen production².

Revised; Unlike other vertebrates, mammalian species possess two types of Leydig cells, fetal-type Leydig cells (FLCs) and adult-type Leydig cells (ALCs), the former of which play pivotal roles in the masculinization of fetuses through androgen production²⁻⁴.

According to the reviewer’s suggestion, the reference (a review paper was cited in the original manuscript) was replaced by the original papers below.

references

- 2 Roosen-Runge, E. C. & Anderson, D. The development of the interstitial cells in the testis of the albino rat. *Acta anatomica* **37**, 125-137, doi: 10.1159/000141460 (1959).

- 3 Miyabayashi, K. *et al.* Alterations in Fetal Leydig Cell Gene Expression during Fetal and Adult Development. *Sexual development* **11**, 53-63, doi:10.1159/000453323 (2017).
- 4 Sararols, P. *et al.* Specific Transcriptomic Signatures and Dual Regulation of Steroidogenesis Between Fetal and Adult Mouse Leydig Cells. *Frontiers in cell and developmental biology* **9**, 695546, doi:10.3389/fcell.2021.695546 (2021).

Reviewer #2: *1. Throughout the manuscript, the authors use the word "progenitors" to describe FLCs that are not differentiated. However, progenitor is a generic term and often leads to confusion. Is it used here to indicate stem Leydig cells of the fetal population? Or FLCs that have begun, but not completed, their differentiation process (stem, progenitor, partly differentiated)?*

We thank the reviewer for indicating the wording issue. In general, it has been accepted that stem cells can be classified into a few groups such as pluripotent, multipotent, and unipotent stem cells based on the potentiality. According to the classification of the stem cells, it seems that developing fetal testes contain unipotent stem cells for FLCs in the interstitial space. Therefore, the cells we found in this study may be stem FLCs. As another important characteristic of the stem cells, it has been accepted that they have a potential to self-renew. Unfortunately, however, we do not have any information whether the cells we found have the potential to self-renew. Therefore, we hesitate to describe the cells as stem FLCs.

As indicated by the reviewer, the term ‘progenitor cells (progenitors)’ may be generic and thus often leads to confusion. However, the term has been used to describe transient stage of cells which are differentiating but not fully differentiated yet. Indeed, many papers using the term were published so far¹⁻⁵. Based on these reasons, we thought that ‘progenitor cells (progenitors)’ might be appropriate for the cells that we currently found.

references

- 1 Akashi, K., Traver, D., Miyamoto, T. & Weissman, I. L. A clonogenic common

- myeloid progenitor that gives rise to all myeloid lineages. *Nature* **404**, 193-197, doi:10.1038/35004599 (2000).
- 2 Domian, I. J. *et al.* Generation of functional ventricular heart muscle from mouse ventricular progenitor cells. *Science* **326**, 426-429, doi:10.1126/science.1177350 (2009).
 - 3 Kumar, M. E. *et al.* Mesenchymal cells. Defining a mesenchymal progenitor niche at single-cell resolution. *Science* **346**, 1258810, doi:10.1126/science.1258810 (2014).
 - 4 Paul, F. *et al.* Transcriptional Heterogeneity and Lineage Commitment in Myeloid Progenitors. *Cell* **163**, 1663-1677, doi:10.1016/j.cell.2015.11.013 (2015).
 - 5 Sanchez-Ferras, O. *et al.* A coordinated progression of progenitor cell states initiates urinary tract development. *Nature communications* **12**, 2627, doi:10.1038/s41467-021-22931-5 (2021).

2. The authors conclude that Tmsb10 is key for Leydig cell differentiation, as mentioned in the title of the manuscript. Although a strong possibility, this conclusion is entirely based on in vitro differentiation/reconstruction assays. Another experimental approach, such as knocking out the Tmsb10 gene, would strengthen this conclusion. The authors have used gene editing to knock-in a mCherry cDNA in frame with the Tmsb10 gene in order to determine where/when Tmsb10 is expressed. With the gene editing approach established (guides, homology arms, constructs), this approach could be easily adapted to knockout the Tmsb10 gene.

We agree with the reviewer that studies with *Tmsb10* knockout (KO) mice could strengthen the results of our *in vitro* studies. Indeed, we attempted to generate the *Tmsb10* KO mouse. Unfortunately, however, we could not obtain *Tmsb10* heterozygous KO mice even though we generated more than 150 genome-edited mice using three different gRNAs. Moreover, when pregnant females were examined at 16.5 dpc, we could not find any dead and *Tmsb10* heterozygous KO embryos. From these results, we currently assume that *Tmsb10* heterozygous KO mice may be embryonic lethal at the

early stage. To solve this issue, we are planning to establish cell- or time-specific *Tmsb10* conditional KO mice. Unfortunately, however, it should take more than one year.

3. Lines 99-107: it seems that this section would be more appropriate for the Introduction section. It is all previously published information.

We thank the reviewer for the comment. According to the suggestion by the reviewer, the section was transferred in the Introduction of the revised manuscript (Page 6, Line 90-98).

4. Lines 101-102: S-EGFP are defined as FLCs. However, W-EGFP are identified as interstitial cells. The interstitium contains several cell types, including stem FLCs and stem ALCs. Could the large number of W-EGFP cells correspond to various interstitial cell types and not only "precursor" FLCs?

As mentioned by the reviewer, the interstitial cells of the fetal testis have been thought to contain several cells at different developmental stages of FLCs and ALCs. Therefore, we expected that the presence of these cells could be revealed by the single cell transcriptome study of W-EGFP cells. In fact, our study showed that W-EGFP cells were divided into three clusters, clusters A, B, and C. Among them, *Tmsb10*-positive cluster C was found to be FLC progenitors. Unexpectedly, however, our study could not provide any evidence that the cells in cluster A or B correspond to stem FLCs or stem ALCs.

5. Since some Tmsb10 positive cells are endothelial cells (co-localize with laminin), is it possible that the W-EGFP could represent late differentiating FLCs from endothelial

origin while the S-EGFP would have a different origin and have differentiated a bit earlier? FLCs are known to have different origins.

We thank the reviewer for the interesting question. As mentioned by the reviewer, FLCs were reported to have two different origins, epithelial and perivascular cells¹, and our study demonstrated that *Tmsb10* is highly expressed in laminin-positive endothelial cells. Taken together, these results may suggest that *Tmsb10*-positive FLC progenitors are derived from *Tmsb10*-positive endothelial cells. Unfortunately, however, our present analysis did not provide any direct evidence to indicate that the *Tmsb10*-positive progenitors are originated from the endothelial cells.

reference

- 1 Kumar, D. L. & DeFalco, T. A perivascular niche for multipotent progenitors in the fetal testis. *Nature communications* **9**, 4519, doi:10.1038/s41467-018-06996-3 (2018).

6. Do W-EGFP cells express classical steroidogenic markers (of FLCs and/or ALCs) when they become S-EGFP in the in vitro reconstruction/differentiation system?

The immunostaining images below are Fig. 1 P-R of a published paper from our group¹. The reconstructed tissues with W-EGFP cells were subjected to double immunostaining for **(P)** EGFP (green) and HSD3B (red), **(Q)** EGFP (green) and Ad4BP/SF-1 (red), or **(R)** EGFP (green) and HSD17B3 (red). The EGFP-positive cells were positive for Leydig cell markers, HSD3B and Ad4BP/SF-1, whereas they were negative for an ALC marker, HSD17B3, indicating that the Leydig cells differentiated in the reconstructed tissues are FLCs.

reference

- 1 Inoue, M. *et al.* Isolation and Characterization of Fetal Leydig Progenitor Cells of Male Mice. *Endocrinology* **157**, 1222-1233, doi:10.1210/en.2015-1773 (2016).

7. The authors integrate the PDGF signalling pathway and showed that PDGF-AA increases pERK probably via the RAS pathway. Does PDGF-AA have any effect on Tmsb10 expression?

According to the comment, we examined whether PDGF-AA affects the expression of *Tmsb10* by qRT-PCR. The W-EGFP cells were incubated in the absence (open bar) or presence of PDGF-AA (green bar). The result indicated that PDGF-AA did not affect the expression of *Tmsb10*.

8. To this reviewer, there are other possibilities to explain the crosstalk between the PDGF, DHH, and possibly the NOTCH pathway (which is not really considered by the authors). NOTCH is known to keep Leydig cells in an undifferentiated state. In addition, there are evidence that the NOTCH and RAS pathways often intersect, either in a synergistic or antagonistic manner. Have the authors considered that NOTCH could work with the RAS pathway to increase pERK maintaining FLCs in an undifferentiated state? At least, the various options (involvement of NOTCH, role of PDGF on Tmsb10 expression, alternate pathways) should be discussed.

We thank the reviewer for the comment. As mentioned by the reviewer, it has been reported that Ras and Notch signaling pathways interact positively or negatively. Moreover, because NOTCH is known to be involved in FLC differentiation, it is possible to assume that NOTCH gives effects on the crosstalk of PDGF and DHH. Therefore, we added the sentences below to the revised manuscript (Page 23, Line 407-Page 24, Line 410).

Moreover, it has been known that NOTCH is involved in FLC differentiation^{7,19} and the molecule cross-talks with Ras pathway⁶⁸. These findings suggest that NOTCH signaling may also be involved in regulation of ciliogenesis.

references

- 7 Tang, H. *et al.* Notch signaling maintains Leydig progenitor cells in the mouse testis. *Development* **135**, 3745-3753, doi:10.1242/dev.024786 (2008).
- 19 Kumar, D. L. & DeFalco, T. A perivascular niche for multipotent progenitors in the fetal testis. *Nature communications* **9**, 4519, doi:10.1038/s41467-018-06996-3 (2018).
- 68 Sundaram, M. V. The love-hate relationship between Ras and Notch. *Genes & development* **19**, 1825-1839, doi:10.1101/gad.1330605 (2005).

Reviewer #3:

1. In Figure 3C-G, images are blurry and appears to have different magnifications

making it hard to draw any conclusions.

We thank the reviewer for the indication. Because Fig. 3C, E, and G in the original manuscript were unclear, these photographs were substituted by bright field photographs in the revised manuscript. Scale bars are added to them.

2. Fig 4A, the immunostaining results appears to be nonspecific and this non specificity is reflected in Fig 4D where, siT10 dose incremental treatment induced non-significant changes in the DHH signaling.

According to the concern raised by the reviewer, we attempted to stain primary cilia with another marker protein, smoothed (SMO), which accumulates in primary cilia upon SAG treatment¹. The immunofluorescence images below are W-EGFP cells double-stained for ARL13B (red) and SMO (white, mouse smoothed monoclonal antibody (Santa Cruz Biotechnology, sc-166685)). As indicated by the arrows, we detected co-localization of ARL13B and SMO. Based on these results, we reason that the signals in the immunostaining images in Fig. 4A were not non-specific. We have not added this figure to the revised version. However, if the reviewer tells that this figure is required, we are willing to add it.

reference

- 1 Rohatgi, R., Milenkovic, L. & Scott, M. P. Patched1 regulates hedgehog signaling at the primary cilium. *Science* **317**, 372-376, doi:10.1126/science.1139740 (2007).

3. In Fig 4H authors depicted an illustration of potential mechanisms of FLC differentiation. However, the results in Fig 4J contradicts the proposed mechanism as inhibiting RAS should not increase the expression of GLI.

Because of the busy and complex figures, the reviewer might be confused. As shown in Fig. 4H (left Fig. below), we assumed that RAS suppresses ciliation and then *Gli1* expression through activating RAF/MEK/ERK. Being consistent with it, inhibition of *Ras* (si*Ras* in Fig. 4J, blue bar in the figure below) resulted in the increase of *Gli1* mRNA.

4. It is unclear why in Fig 4F, authors compared the impact of presence or absence of SAG on pERK in context of si*T10* when phosphorylation of ERK according to the illustration in Fig 4h is upstream of DHH signaling?

As indicated by the reviewer, we assume that phosphorylation of ERK is localized at the upstream of DHH signaling. In addition to this regulation, it has been reported that hedgehog signal activates RAS/ERK pathway in several cell types^{46,47}. In order to depict a hypothetical Fig. 4H, we wanted to exclude the possibility that DHH activates

RAS/ERK pathway in W-EGFP cells. Therefore, si*T10* or si*Cnt*-treated W-EGFP cells were further treated with or without SAG, and then the phosphorylation levels of ERK were examined. Expectedly, the results shown in Fig. 4F indicate that hedgehog signaling does not affect the RAS/ERK pathway in W-EGFP cells.

To avoid confusion of the potential readers, we revised the text below to describe why we examined phosphorylation of ERK in the presence of SAG in the revised version (Page14, Line 228-234 in the revised manuscript).

Original; The amount of phosphorylated ERK (pERK) was increased by *Tmsb10* KD both in the presence and absence of SAG, whereas the amount of ERK was unaffected (Fig. 4f, g).

Revised; The amount of phosphorylated ERK (pERK) was increased by *Tmsb10* KD in the absence of SAG (Fig. 4f, g). A few papers reported that hedgehog signal activates RAS/ERK pathway in several cell types^{46,47}. To exclude the possibility that DHH signal activates RAS/ERK pathway in W-EGFP cells, we examined whether pERK is affected by SAG treatment. As the result, we found that the amount of pERK was not changed by the treatment, strongly suggesting that DHH signal does not affect RAS/ERK pathway in W-EGFP cells.

references

- 46 Osawa, H. *et al.* Sonic hedgehog stimulates the proliferation of rat gastric mucosal cells through ERK activation by elevating intracellular calcium concentration. *Biochemical and biophysical research communications* **344**, 680-687, doi:10.1016/j.bbrc.2006.03.188 (2006).
- 47 Rovidia, E. & Stecca, B. Mitogen-activated protein kinases and Hedgehog-GLI signaling in cancer: A crosstalk providing therapeutic opportunities? *Seminars in cancer biology* **35**, 154-167, doi:10.1016/j.semcancer.2015.08.003 (2015).

5. Similarly, It is unclear why in Fig 5C, authors compared the impact of presence or absence of SAG on pERK in context of PDGF when phosphorylation of ERK according

to the illustration in Fig 4H and 5G is upstream of DHH signaling?

We thank the reviewer for the comment. As we mentioned in our answer to the comment 4 above, we examined the possibility again whether SAG treatment affects ERK phosphorylation. However, because this possibility was excluded by the study shown in Fig. 4f and g, the left lane (SAG+, PDGF-) of Fig. 5c and the bar of Fig. 5d (SAG+, PDGF-) of the original version (figure at the down left) is deleted from the Fig. 5c of the revised version (figure at the down right). Accordingly, a sentence ‘SAG treatment had no effect on the amount of pERK’ at Page 15, Line 262-263 in the original version was deleted from the revised version.

6. Importantly, authors should perform gain and loss of function experiments to validate the proposed mechanism in illustrations 4H, 5G and 6A in context of each of the candidates included.

We agree with the notion that gain-of-function studies could strongly support our conclusion obtained from the loss-of-function studies. Therefore, we attempted to overexpress *Tmsb10* in W-EGFP cells by using multiple lipofection reagents, Lipofectamine 2000 (Thermo Fisher Scientific, Waltham, MA, USA), X-tremeGENE HP (Roche Diagnostics Corp., Indianapolis, IN, USA), transIT-2020 (Takara, Shiga, Japan), and FuGENE HD (Promega, Madison, WI, USA).

First, we examined the transfection efficiency. An expression plasmid for a red fluorescent protein, ptd-Tomato-N1, was transfected to W-EGFP and HEK293 cells under several conditions. As shown in the figure below, ptd-Tomato-N1 fluorescence

was clearly observed in the HEK293 cells for 48 h after the transfection. Transfection efficiencies were more than 70%, no matter which reagent was used (summarized in the table below). Unfortunately, however, only a small number of the fluorescence-positive W-EGFP cells were detected after the transfection. The highest transfection efficiency was only about 13% (FuGENE HD Transfection Reagent, DNA: 200 ng, reagent: 0.6 μ l).

		Tomato-positive cells (%)	
X-tremeGENE HP (ul)	DNA (ng)	HEK293	W-EGFP
0.2	100	30.5	2.7
	200	28.8	0.6
0.4	100	26.2	2.3
	200	86.8	7.9

transT-2020 (ul)			
0.45	100	24.8	7.9
	200	74.8	9.1
0.9	100	31.8	1.8
	200	71.9	4.7

FuGENE HD (ul)			
0.6	100	91.3	13.2
	200	94.3	13.4
0.9	100	73.8	11.4
	200	90.1	7.3

Next, we investigated whether *Tmsb10* overexpression affects *Gli1* expression and FLC differentiation. W-EGFP cells were transfected with an expression plasmid, pHA-Tmsb10, under the optimal condition obtained with FuGENE HD. Then, the W-EGFP cells transfected with HA-TMSB10 or HA (control) were used for qRT-PCR and *in vitro* testis reconstruction assays. As expected, *Gli* expression (left graph below) and FLC differentiation (right graph below) were statistically upregulated by *Tmsb10* overexpression, although the increase was only 1.1- and 1.2-fold, respectively. We assume that the weak effects of *Tmsb10* overexpression on *Gli* expression and FLC differentiation are attributable to the low transfection efficiency of the expression plasmid into W-EGFP cells.

We also attempted to overexpress *Tmsb10* using an adenoviruses plasmid containing *Tmsb10*-IRES2-DsRed. Two lots of adenoviruses carrying the plasmid were prepared and they were infected to W-EGFP and HEK293 cells. DsRed fluorescence was detected in the HEK293 cells 48 h after the infection, and the efficiencies were more than 50%. Unexpectedly, however, no signal was detected in W-EGFP cells after the infection.

In our impression, it will take a longer time to establish the method for overexpression of *Tmsb10* in W-EGFP cells. Although we understand the importance of the gain-of-function study, as we described above, it seems currently impossible to include it in our manuscript.

REVIEWERS' COMMENTS:

Reviewer #2 (Remarks to the Author):

The authors have appropriately addressed all my concerns. I do not have other comments.

Reviewer #3 (Remarks to the Author):

The authors have addressed the concerns.